# Engineering a membrane protein chaperone to ameliorate the proteotoxicity of mutant huntingtin

Jeonghyun Oh [1,5], Christy Catherine [1,5], Eun Seon Kim[2,5], Kwang Wook Min[1,5], Hae Chan Jeong[2], Hyojin Kim [1], Mijin Kim[1], Seung Hae Ahn[1], Nataliia Lukianenko [3], Min Gu Jo[2], Hyeon Seok Bak [1], Sungsu Lim[3], Yun Kyung Kim [3], Ho Min Kim [1,4] ✉, Sung Bae Lee [2] ✉ & Hyunju Cho [1] ✉

Toxic protein aggregates are associated with various neurodegenerative diseases, including Huntington's disease (HD). Since no current treatment delays the progression of HD, we develop a mechanistic approach to prevent mutant huntingtin (mHttex1) aggregation. Here, we engineer the ATP-independent cytosolic chaperone PEX19, which targets peroxisomal membrane proteins to peroxisomes, to remove mHttex1 aggregates. Using yeast toxicity-based screening with a random mutant library, we identify two yeast PEX19 variants and engineer equivalent mutations into human PEX19 (hsPEX19). These variants effectively delay mHttex1 aggregation in vitro and in cellular HD models. The mutated hydrophobic residue in the α4 helix of hsPEX19 variants binds to the N17 domain of mHttex1, thereby inhibiting the initial aggregation process. Overexpression of the hsPEX19-FV variant rescues HD-associated phenotypes in primary striatal neurons and in Drosophila. Overall, our data reveal that engineering ATP-independent membrane protein chaperones is a promising therapeutic approach for rational targeting of mHttex1 aggregation in HD.

Maintaining proper protein homeostasis is essential for healthy cells. However, the cell is under continuous risk from newly synthesized proteins that might expose hydrophobic surfaces in the crowded cellular environment, leading to protein misfolding and aggregation[1-4]. To overcome these problems, cells invest in a sophisticated integrative chaperone network that supports accurate de novo protein folding, facilitates refolding of misfolded proteins, and prevents protein aggregation[1,2,5,6]. However, environmental stresses, genetic mutations, and aging can reduce the overall capacity of molecular chaperones, resulting in the accumulation of toxic aggregates and misfolded proteins in cells[7-9]. Such aggregates eventually lead to various diseases, including neurodegenerative diseases and type 2 diabetes[10,11].

Huntington's disease (HD) is the most common dominantly inherited neurodegenerative disorder and is caused by the abnormal expansion of CAG (polyQ) repeats in exon 1 of the huntingtin gene (Httex1)[12,13]. The length of polyQ repeats in the mutant Httex1 (mHttex1 with > 36 repeats) positively correlates with an increasing propensity to form aggregates and correlates inversely with the age of disease onset[14,15]. Aggregation of the polyQ repeat domain is also mediated by its flanking domains: the N-terminal conserved N17 domain and the C-terminal proline-rich domain (PRD). The N17 domain stimulates mHttex1 aggregation, whereas the PRD inhibits it[16-19]. Accumulation of mHttex1 aggregates in the nucleus and cytoplasm impairs the proteostasis network and disrupts cellular endomembranes, thus leading

[1]Center for Biomolecular and Cellular Structure, Institute for Basic Science (IBS), Daejeon, Republic of Korea. [2]Department of Brain Sciences, Daegu Gyeongbuk Institute of Science and Technology (DGIST), Daegu, Republic of Korea. [3]Center for Brain Disorders, Brain Science Institute, Korea Institute of Science and Technology (KIST), Seoul, Republic of Korea. [4]Department of Biological Sciences, Korea Advanced Institute of Science and Technology (KAIST), Daejeon, Republic of Korea. [5]These authors contributed equally: Jeonghyun Oh, Christy Catherine, Eun Seon Kim, Kwang Wook Min. ✉e-mail: hm_kim@kaist.ac.kr; sblee@dgist.ac.kr; hjcho@ibs.re.kr

to dysregulation of diverse cellular processes including transcription, mitochondrial respiration, ER homeostasis, vesicular trafficking, and axonal transport[20–23].

One suggested approach to correcting protein misfolding and removing pathological aggregates involves engineering a molecular chaperone to increase chaperone capacity in affected cells[24–26]. Indeed, the yeast AAA + protein disaggregase, Hsp104, has been engineered to rescue the proteotoxicity of TDP43, FUS, and α-synuclein for amyotrophic lateral sclerosis (ALS) and Parkinson's disease[24,27,28]. However, most chaperones, including Hsp104, require subunit assembly, oligomerization, co-chaperones, or cofactors, such as ATP and metal ions, for optimal activity or substrate specificity[5,29]. Thus, engineered chaperones that rely on cellular ATP concentrations and the expression levels of their subunits and co-chaperones[30] may complicate therapeutic applications. Furthermore, several ATP-independent human disaggregases such as DAXX, Karyopherin-β2, and high-temperature requirement A1 (HTRA1) are known to reverse the formation of various pathological aggregates[31–33]. These studies suggest that ATP-independent chaperones are promising therapeutic targets for treating neurodegenerative diseases.

Peroxisomal dysfunction in peroxisome biogenesis disorders is linked to neurodegenerative diseases[34–37]. Intriguingly, a previous study showed that *PEX19−/−* adult flies exhibit a defect in climbing ability, suggesting the potential functional relevance of PEX19 to neurodegeneration[38]. PEX19, an ATP-independent cytosolic chaperone, mediates the targeting of peroxisomal membrane proteins (PMPs) during peroxisome biogenesis[39–41]. Importantly, PEX19 does not require any co-chaperones, cofactors, or complex assembly steps for its chaperone activity. Therefore, we hypothesize that PEX19 could be readily engineered to provide a robust approach for mitigating mHttex1 proteotoxicity.

Here, using yeast toxicity-based screening[42] with a random mutant library, we isolate yeast PEX19 (*sc*PEX19) variants that suppress the proteotoxicity of mHttex1 aggregates. Using this information, we engineer the equivalent human PEX19 (*hs*PEX19) variants and show that they also potently suppress toxic mHttex1 aggregates. The isolated *hs*PEX19 variants directly bind the hydrophobic side of the amphipathic helix at the N17 domain of mHttex1, thereby effectively delaying the kinetics of mHttex1 aggregation. Overexpression of the *hs*PEX19 variant further rescues mHttex1-induced neurite degeneration in mouse striatal neurons and improves both the climbing ability and lifespan of flies expressing mHttex1-93Q. Altogether, our study suggests that fine-tuning the sequences of ATP-independent membrane protein chaperones could be a feasible approach to designing therapeutic chaperones for HD and potentially other diseases linked to protein aggregation.

## Results

### Engineered *sc*PEX19 variants suppress the toxicity of mHttex1 in yeast

To isolate a *sc*PEX19 mutant gene that suppresses the cellular toxicity of mHttex1 protein, we used the yeast toxicity-based screening method[42] (Fig. 1a). Deletion of PRD in Httex1-97Q enhances its polyQ-induced toxicity in yeast[19]. This mutant is more optimal for screening since it results in a larger difference in cell viability compared to expressing non-toxic Httex1-25Q. To this end, we generated yeast strains carrying chromosomally integrated Httex1 genes (Httex1-25QΔP and Httex1-97QΔP), which encode an N-terminal FLAG tag, the first 17 amino acids of Httex1 (N17 domain), 25 or 97 repeats of glutamine, and a C-terminal GFP gene under the control of the galactose-inducible promoter (Fig. 1a). Expression of the wild-type *sc*PEX19 did not alter the cellular toxicity of Httex1-97QΔP when compared with the empty vector control (Supplementary Fig. 1a). We randomly mutated the entire *sc*PEX19 gene and screened the *sc*PEX19 plasmid library (~ $2.5 \times 10^5$ library size) against Httex1-97QΔP toxicity. Among approximately 90,000 transformants, 21 colonies were able to grow on galactose plates. After assessing the cell

viability of those colonies, we found that two colonies containing *sc*PEX19 variants, m1 and m2, effectively suppressed the cellular toxicity of Httex1-97QΔP in yeast (Fig. 1b). Sequencing analysis based on colony PCR showed that these two *sc*PEX19 variants displayed distinct single chromatogram peaks at the mutation sites (Supplementary Fig. 2). Thus, although yeast cells often harbor multiple plasmids, the selected colonies are highly likely to contain a single plasmid with the identified multiple mutations listed in Fig. 1a.

The isolated *sc*PEX19 variants share two common mutation sites, L288F and E292V (Fig. 1a). Therefore, we hypothesized that these two mutation sites account for the ability of *sc*PEX19 variants to rescue Httex1-97QΔP-induced toxicity in yeast. To test this hypothesis, we generated a double mutant *sc*PEX19-L288F/E292V. The results of the spotting assay showed that *sc*PEX19-L288F/E292V is sufficient to suppress the cellular toxicity of Httex1-97QΔP (Fig. 1b). In contrast, coexpression of the single mutants of *sc*PEX19-L288F or *sc*PEX19-E292V with Httex1-97QΔP did not restore cell viability (Supplementary Fig. 1b), suggesting that *sc*PEX19-L288F/E292V is a minimally mutated suppressor of polyQ-induced toxicity in yeast. In addition, we substituted E292 with other hydrophobic amino acids on the *sc*PEX19 variant. We found that only *sc*PEX19-L288F/E292I suppressed Httex1-97QΔP-induced toxicity to the same degree as *sc*PEX19-L288F/E292V (Fig. 1c), possibly due to the structural similarity between the valine and isoleucine side chains. Moreover, overexpression of *sc*PEX19-L288F/E292V or *sc*PEX19-L288F/E292I alone did not cause cellular toxicity at either 30 °C or 37 °C (Supplementary Fig. 1c). Therefore, we identified two *sc*PEX19 variants, *sc*PEX19-L288F/E292V (*sc*PEX19-FV) and *sc*PEX19-L288F/E292I (*sc*PEX19-FI), that potently suppress mHttex1 toxicity in yeast.

Consistent with the results obtained with the spotting assay, both microscopy and Western blot analyses showed that *sc*PEX19-FV and *sc*PEX19-FI significantly reduced the aggregation of Httex1-97QΔP proteins compared to *sc*PEX19-WT (Fig. 1d–g). Over 50% of Httex1-97QΔP was found in SDS-insoluble aggregates in *sc*PEX19-WT expressing cells (Fig. 1f, g). In contrast, overexpression of *sc*PEX19-FV and *sc*PEX19-FI drastically reduced the relative amount of SDS-insoluble 97Q aggregate and simultaneously increased SDS-soluble 97Q protein levels (Fig. 1f, g, and Supplementary Fig. 1d–f). This enhancement of Httex1-97QΔP solubility by *sc*PEX19-FV and *sc*PEX19-FI is not due to different expression levels of PEX19 in the cells (Supplementary Fig. 1d, g).

During protein targeting to peroxisome membranes, farnesylation of the C-terminal cysteine residue in the PEX19-CaaX motif increases the binding affinity of PMPs, effectively preventing PMP aggregation in the cytosol[43]. In addition, the PMP-bound PEX19 is recruited to the peroxisomal membrane by PEX3, which interacts with an N-terminal αα helix in PEX19[44–46]. Noticeably, the levels of farnesylated PEX19 were significantly reduced in *sc*PEX19-FV and *sc*PEX19-FI overexpressing cells (Supplementary Fig. 1d, h). Therefore, we tested whether two major aspects of PEX19-mediated PMP targeting, farnesylation and interaction with PEX3[43–45,47], are crucial for ameliorating polyQ-induced cellular toxicity in yeast. To this end, we introduced two further mutations, a farnesylation-defective mutation *sc*PEX19-C339S and a PEX3 binding-defective mutation *sc*PEX19-ΔN, into *sc*PEX19-WT and the toxicity-reducing variants (Supplementary Fig. 3a). The results of a spotting assay showed that coexpression of *sc*PEX19-FV/C339S, *sc*PEX19-FV/ΔN, *sc*PEX19-FI/C339S, and *sc*PEX19-FI/ΔN with Httex1-97QΔP did not alter cell growth compared to *sc*PEX19-FV and *sc*PEX19-FI (Supplementary Fig. 3b, c). Therefore, both farnesylation and recruitment of PEX19 to the peroxisomal membrane by binding to PEX3 are dispensable for *sc*PEX19-FV and *sc*PEX19-FI to suppress the cellular toxicity of Httex1-97QΔP in yeast.

### *hs*PEX19 variants suppress mHttex1 aggregation

Due to the immunogenic potential of yeast gene-derived proteins, we used the *sc*PEX19 variants to create equivalent human PEX19 (*hs*PEX19)

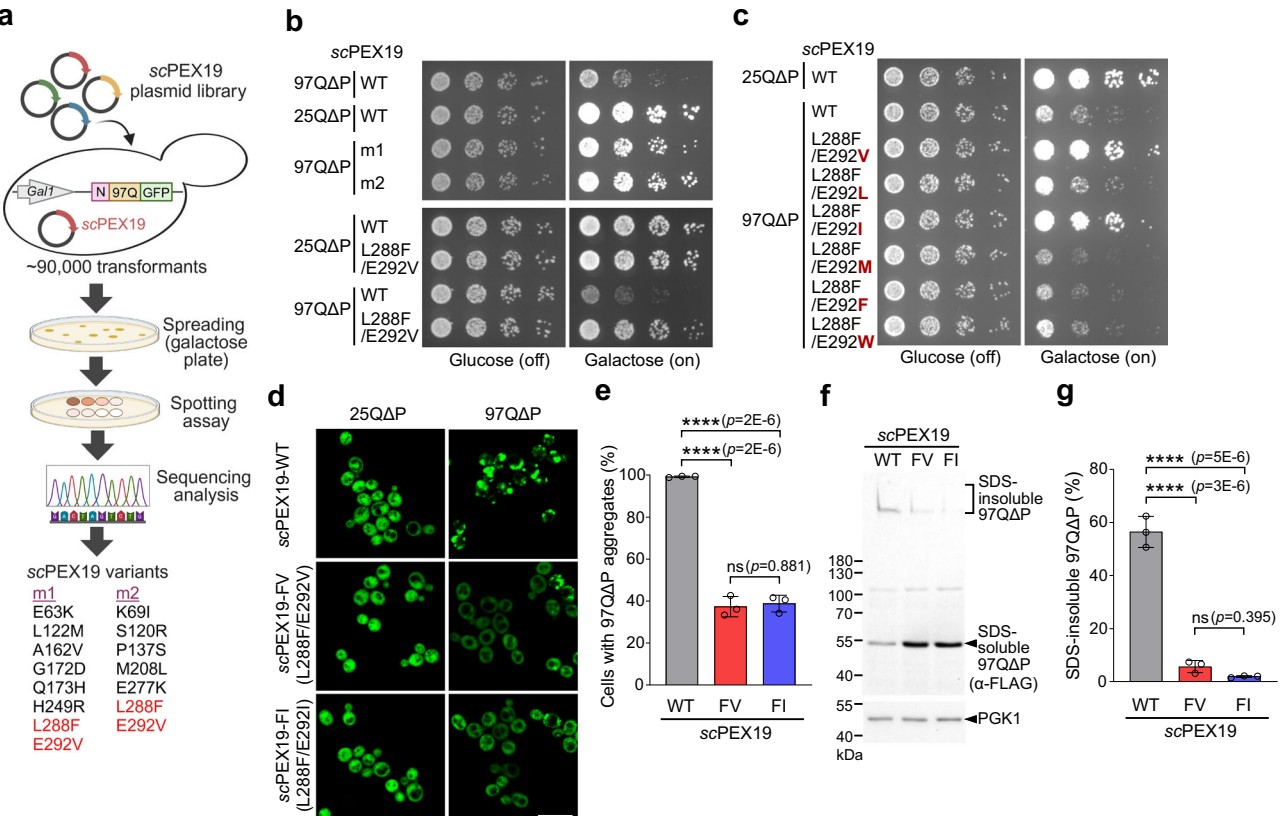

**Fig. 1 | Identification of *sc*PEX19 variants that suppress cellular toxicity of mHttex1. a** Yeast toxicity-based screening to identify mHttex1 suppressors. *sc*PEX19 plasmid library generated by random mutagenesis was transformed into Httex1-97QΔP-GFP-integrated yeast cells. Both *sc*PEX19 variants and Httex1-97QΔP-GFP are under the control of the *GAL1* promoter. The sequences of identified *sc*PEX19 mutants, m1, and m2, are shown, and the common mutation sites are highlighted in red. Created in BioRender. Hyunju, C. (2025) https://BioRender.com/o38y171. **b, c** Growth test of Httex1-25QΔP-GFP- and Httex1-97QΔP-GFP-integrated yeast cells expressing *sc*PEX19-WT and its *sc*PEX19 variants. Five-fold serial dilutions of cells were spotted on galactose plates to coexpress Httex1-25QΔP or Httex1-97QΔP and the indicated *sc*PEX19 proteins (Right) or on glucose plates as loading controls (Left). Representative images from three biological replicates (*n* = 3). **d, e (d)** Confocal microscopy images of Httex1-25QΔP-GFP and Httex1-97QΔP-GFP cells upon coexpression of *sc*PEX19-WT, *sc*PEX1.9-FV, and *sc*PEX19-FI. Scale bar: 10 μm. The percentage of cells containing 97QΔP aggregates was quantified in (**e**). **f, g (f)** Representative image of Western blot monitoring SDS-insoluble and SDS-soluble Httex1-97QΔP-GFP proteins and (**g**) quantification of SDS-insoluble protein in (**f**). Yeast cells were lysed using glass beads, and then the total cell lysates were analyzed using Western blot. N-terminally FLAG-tagged Httex1-97QΔP-GFP was probed using FLAG antibody. PGK1 serves as a loading control. Data in (**e**), (**g**) are shown as mean ± SD, with three biological replicates (*n* = 3). Pairwise comparisons are shown as indicated, where ****$p < 0.0001$ by the ordinary one-way ANOVA with Tukey post-hoc test. Source data are provided as a Source Data file.

variants. Sequence alignment analysis showed that both mutated residues (L288F/E292V and L288F/E292I) are located in the α4 helix of the PEX19 protein (Fig. 2a, green highlighted boxes). These residues are highly conserved from Human (M255/Q259) to Arabidopsis (M202/Q206) (Fig. 2a). In addition, the M255 residue of *hs*PEX19 directly interacts with the farnesyl group near its C-terminal end[43] (Fig. 2b), suggesting that this residue could be important for substrate recognition. Due to their highly homologous sequences, we hypothesized that introducing identical mutations (M255F/Q259V or M255F/Q259I) into *hs*PEX19 could also enhance the suppression of mHttex1 aggregation.

To test whether both purified *sc*PEX19 and *hs*PEX19 variants directly prevent Httex1-51Q aggregation in vitro, we used the well-established filter trap assay that detects heat-stable, SDS-insoluble aggregates[16,48]. In this assay, the N-terminal Httex1-51Q can be exposed by cleaving off a GST-tag using TEV protease, thus initiating polyQ aggregation (Supplementary Fig. 4a). In the absence of a chaperone, Httex1-51Q readily formed SDS-insoluble aggregates at 3 h (Fig. 2c, d and Supplementary Fig. 4b, c). In contrast, equimolar addition of *sc*PEX19-FV and *sc*PEX19-FI completely suppressed aggregation of the purified Httex1-51Q protein, while *sc*PEX19-WT was insufficient to prevent Httex1-51Q aggregation (Supplementary

Fig. 4b, c). Similar to *sc*PEX19 variants, *hs*PEX19 variants effectively delayed aggregation of Httex1-51Q protein in vitro although *hs*PEX19-FI exhibited weaker chaperone activity for Httex1-51Q than *hs*PEX19-FV (Fig. 2c, d). This enhanced chaperone activity of *hs*PEX19 variants was not due to different TEV cleavage efficiency caused by their mutations (Supplementary Fig. 4d). Furthermore, *hs*PEX19-FV suppressed aggregation of Httex1-51Q more effectively than the known Httex1 aggregation suppressor, nascent polypeptide-associated complex (NAC)[49] (Supplementary Fig. 4e–g).

Both the ThioflavinT assay and negatively stained transmission electron micrograph (TEM) analysis showed that *hs*PEX19-FV effectively suppressed the formation of Httex1-51Q fibrils (Fig. 2e, f, and Supplementary Fig. 5a, b). In contrast, *hs*PEX19-WT did not prevent Httex1-51Q fibril formation (Fig. 2e, f, and Supplementary Fig. 5a, b). Consistent with the results of the filter trap assay (Fig. 2c, d, and Supplementary Fig. 5c, d), *hs*PEX19-FI also delayed fibril formation of Httex1-51Q (Fig. 2e), albeit less effectively than *hs*PEX19-FV. TEM analysis of *hs*PEX19-FI revealed both larger Httex1-51Q aggregates and small fibrils at 15 h and 24 h, while *hs*PEX19-FV exhibited a complete suppression of Httex1-51Q fibril formation at the tested time points (Fig. 2f and Supplementary Fig. 5a, b). Collectively, *hs*PEX19-FV

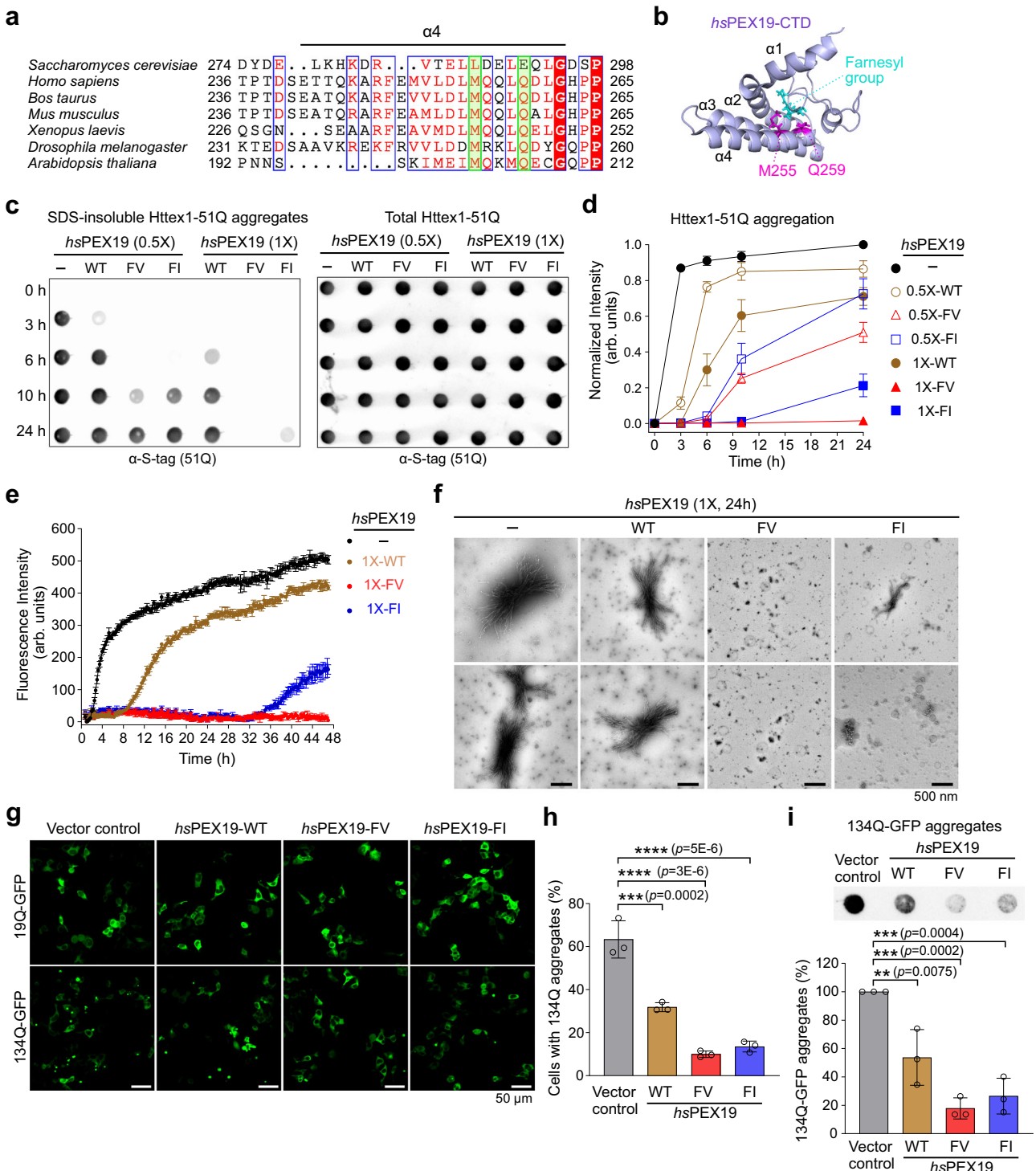

Fig. 2 | *hs*PEX19 variants suppress mHttex1 aggregation in vitro and in mammalian cells. a Multiple sequence alignment of the α4 helix sequences of PEX19 across various species. The alignment was performed using the Clustal Omega and displayed with ESPript 3[101,102]. Conserved sequences of *sc*PEX19-L288 and *sc*PEX19-E292 are highlighted as green boxes. b NMR structure of *hs*PEX19-CTD (161–299 aa) (PDB 5LNF)[43]. Two conserved residues, M255 and Q259, are located in the α4 helix of *hs*PEX19 and are shown in magenta. The M255 residue of *hs*PEX19 is known to bind its C-terminally modified farnesyl group (cyan)[43]. c, d In vitro aggregation assay of Httex1-51Q in the absence and presence of *hs*PEX19 proteins. 3 µM of GST-TEV-Httex1-51Q-Stag and 1.5 µM (0.5×) or 3 µM (1×) of PEX19 proteins were incubated at 30 °C, and after the addition of TEV protease, samples were quenched at the indicated time points. SDS-insoluble Httex1-51Q aggregates in (c) and their replicates were quantified and shown in (d) (n = 3, mean ± SD). e ThioflavinT fluorescence assay to measure fibril formation of Httex1-51Q. The

fluorescence intensity was measured every 15 min. Data are shown as mean ± SD with n = 3 (technical replicates). f Negatively stained transmission electron micrograph (TEM) images of Httex1-51Q in the absence and presence of *hs*PEX19 proteins. Scale bar: 500 nm g, h (g) Confocal microscopy images of HEK293T cells coexpressing Httex1-19Q-GFP or Httex1-134Q-GFP and *hs*PEX19. Empty vector control (denoted as vector control) was used as a negative control. Scale bar: 50 µm. h The percentage of cells containing 134Q aggregates was quantified. i (top) Representative image of the filter trap assays monitoring the SDS-insoluble Httex1-134Q-GFP aggregates in HEK293T cells upon coexpression of *hs*PEX19 proteins. (bottom) Quantification of the images and their replicates. Data in (h, i) are shown as mean ± SD, with three biological replicates (n = 3). Pairwise comparisons are shown as indicated, where **p < 0.01, ***p < 0.001, ****p < 0.0001 by the ordinary one-way ANOVA with Tukey post-hoc test. Source data are provided as a Source Data file.

suppressed Httex1-51Q aggregation and fibril formation more effectively than *hs*PEX19-FI.

Similar to other amyloid diseases, external seeds are known to promote aggregation of mHttex1 in vitro[50,51]. Furthermore, the mHttex1 seeding activities detected in cerebrospinal fluid and brain tissues of HD mice and patients have been suggested to positively correlate with disease progression[50,52,53]. Therefore, we tested whether *hs*PEX19 variants inhibit the seed-catalyzed aggregation of mHttex1 (Supplementary Fig. 6a, b). In the presence of 100 nM of seed (~ 3% of total Httex1-51Q), fibril formation of Httex1-51Q was significantly increased compared to the non-seeded sample at 36 h (Supplementary Fig. 6a). Both *hs*PEX19-FV and *hs*PEX19-FI exhibited a complete suppression of Httex1-51Q fibril formation at 36 h, even in the presence of 100 nM seed (Supplementary Fig. 6b), suggesting that the PEX19 variants exhibit the potential to reduce the seeded aggregation of mHttex1. In contrast, *hs*PEX19 variants were unable to redissolve preformed Httex1-51Q aggregates when added at 3 h, suggesting that they do not have disaggregase activity (Supplementary Fig. 6c, d). Therefore, we conclude that *hs*PEX19 variants function as a holdase that prevents the initial aggregation process of Httex1-51Q.

To test whether *hs*PEX19 variants are also effective in reducing mHttex1 aggregation in a mammalian HD model, we coexpressed *hs*PEX19 variants with Httex1-19Q-GFP or Httex1-134Q-GFP in HEK293T cells[54]. Overexpression of *hs*PEX19-FV and *hs*PEX19-FI at ~ 3-fold over endogenous PEX19 levels strongly prevented the aggregation of Httex1-134Q, as demonstrated by both fluorescence microscopy analysis and the filter trap assay (Fig. 2g–i and Supplementary Fig. 7a–d). In contrast, overexpression of *hs*PEX19-WT reduced the Httex1-134Q aggregates by ~ 50% on average, suggesting that *hs*PEX19-WT itself exhibits a mild chaperone activity toward mHttex1 proteins, as also supported by the in vitro aggregation assays (Fig. 2g–i). The difference in rescuing effects observed in *hs*PEX19 variants relative to their wild-type protein was not due to different expression levels of exogenous PEX19 or Httex1-134Q (Supplementary Fig. 7a–c). In contrast to *sc*PEX19 variants, over 95% of both *hs*PEX19-WT and its variants were farnesylated in HEK293T cells (Supplementary Fig. 7d). Consistent with this, overexpression of *hs*PEX19 variants did not perturb the peroxisomal localization of the peroxisomal membrane protein PMP70, suggesting that this approach is unlikely to interfere with peroxisome biogenesis (Supplementary Fig. 7e, f). Therefore, these data demonstrate that the substitution of two conserved residues on the α4 helix of *hs*PEX19 significantly increases its chaperone activity toward mHttex1.

## PEX19 variants bind the N17 domain of mHttex1

The N17 domain of Httex1 has an amphipathic helical property, which contributes to the initiation and acceleration of mHttex1 aggregation[16,55] (Fig. 3a). Furthermore, a recent study suggested that structural coupling between the N17 and polyQ repeat domains stabilizes the helical content of Httex1 and accelerates aggregation[56]. Deletion of the N17 domain of Httex1-51Q (Httex1-51Q-ΔN) delays the kinetics of Httex1-51Q aggregation[16]. Given that *hs*PEX19 variants generate a more hydrophobic environment at their C-terminal domain (CTD) than *hs*PEX19-WT, we hypothesized that they bind to the hydrophobic amino acids in mHttex1, possibly at the N17 domain of mHttex1. Thus, we tested whether *hs*PEX19 variants also suppress Httex1-51Q-ΔN aggregation in vitro. In contrast to Httex1-51Q-WT, *hs*PEX19 variants were insufficient to suppress the aggregation of Httex1-51Q-ΔN (Fig. 3b, c). Consistent with this, GST-Httex1-51Q-WT readily bound to *hs*PEX19-FV, whereas no binding was observed with GST-Httex1-51Q-ΔN (Supplementary Fig. 8a, b). Furthermore, *hs*PEX19 variants did not suppress the aggregation of another polyQ repeat protein, Ataxin3 (Fig. 3d–f). In contrast, both *hs*PEX19-WT and its variants reduced aggregation of the N-terminally N17-fused Ataxin3-78Q (N17-ATX3-78Q) to a similar extent (Fig. 3e, g). Since the N17

domain in N17-Ataxin3-78Q is distally located from the polyQ repeat domain (Fig. 3d), it is plausible that *hs*PEX19 variants more effectively suppress aggregation of the polyQ proteins when N17 is coupled to the polyQ repeat domain. Taken together, these results suggest that the N17 domain could be the primary recognition site of *hs*PEX19 variants within the Httex1-51Q protein.

To check whether the mutated hydrophobic residues in the *hs*PEX19 variants directly interact with Httex1-51Q, we used the Bpa crosslinking assay that employs a photocrosslinker, p-benzoyl-l-phenylalanine (Bpa) (Supplementary Fig. 9a). We site-specifically incorporated Bpa into the F255 residue of the *hs*PEX19 variants using amber suppression[57]. The addition of equimolar concentration of *hs*PEX19-FV[Bpa] and *hs*PEX19-FI[Bpa] to Httex1-51Q suppressed Httex1-51Q aggregation at 3 and 6 h (Supplementary Fig. 9b, c). At 3 h incubation, *hs*PEX19-FV×Httex1-51Q or *hs*PEX19-FI×Httex1-51Q crosslink at ~ 70 kDa was readily detectable, whereas there was no observed crosslinked band in the presence of Httex1-51Q-ΔN (Fig. 3h, i). Therefore, these results indicate that the F255 residue in the *hs*PEX19 variants specifically binds to the N17 domain of Httex1-51Q.

We further tested whether the hydrophobic amino acid residues in the N-terminal amphipathic helix of mHttex1 also bind the *hs*PEX19 variants (Fig. 3a). To minimize structural perturbation, we incorporated Bpa at the F11 residue of Httex1-51Q among seven hydrophobic amino acid residues (Fig. 3a). Similar to Httex1-51Q-WT, both *hs*PEX19-FV and *hs*PEX19-FI suppressed the aggregation of Httex1-51Q-F11[Bpa] more efficiently than *hs*PEX19-WT (Supplementary Fig. 9d, e). In the presence of *hs*PEX19 variants, two distinct *hs*PEX19×Httex1-51Q crosslinked bands at ~ 70 and ~ 80 kDa were observed (Fig. 3j, k), suggesting that Httex1-51Q -F11[Bpa] binds *hs*PEX19 variants, possibly with two different conformations. In contrast, *hs*PEX19-WT resulted in a distinct crosslinked band at ~ 80 kDa and a weak diffuse band at ~ 70 kDa (Fig. 3j, k). Consistent with its mild chaperone activity, *hs*PEX19-WT also binds to Httex1-51Q-F11[Bpa], but likely with one dominant conformation (Fig. 3j, k). These observed differences in the aggregation and Bpa crosslinking assays are not due to different TEV cleavage efficiency (Supplementary Fig. 9f). Therefore, our data demonstrate that the F11 hydrophobic residue on Httex1-51Q directly interacts with *hs*PEX19 and consistent with the results in Fig. 2d, its variants have stronger chaperone activities for mHttex1.

## The α4 helix of *hs*PEX19 variants serves as a specific binding site for the N17 domain of mHttex1

PEX19 binds to the moderately hydrophobic transmembrane domains (TMDs) of peroxisomal and mitochondrial membrane proteins[39,46,58,59] (Fig. 4a). In addition, PEX19 interacts with TMDs located in diverse topologies of membrane proteins, multi-spanning PMPs, tail-anchored membrane proteins (TAs), and N-terminal signal-anchored membrane proteins (Fig. 4a). Since *hs*PEX19 binds to these moderately hydrophobic TMDs, we hypothesized that *hs*PEX19 variants might also interact with the isolated N17 domain of Httex1 (Fig. 4a). To test this, we fused the N17 domain of Httex1 to the N-terminus of the Maltose binding protein (MBP) (Fig. 4b). The *hs*PEX19-FV[Bpa] and *hs*PEX19-FI[Bpa] proteins readily crosslinked to the N17-MBP protein, whereas no crosslinked band appeared in the presence of wild-type MBP protein (Fig. 4c). These results suggest that the N17 domain of Httex1 is a minimum recognition motif for *hs*PEX19 variants that allows suppression of the mHttex1 aggregation.

To test the substrate specificity of *hs*PEX19 variants, we generated three mutants of the N17-MBP protein: N17-sc-MBP, N17-5AG-MBP, and N17-2I-MBP. N17-sc-MBP contains a scrambled sequence of five hydrophobic residues in the N17 domain, thus having no alterations in the hydrophobicity of the N17 domain (Fig. 4a). Replacement of hydrophobic residues in the N17 domain with five alanine/glycines reduces the hydrophobicity of the N17 domain (N17-5AG-MBP),

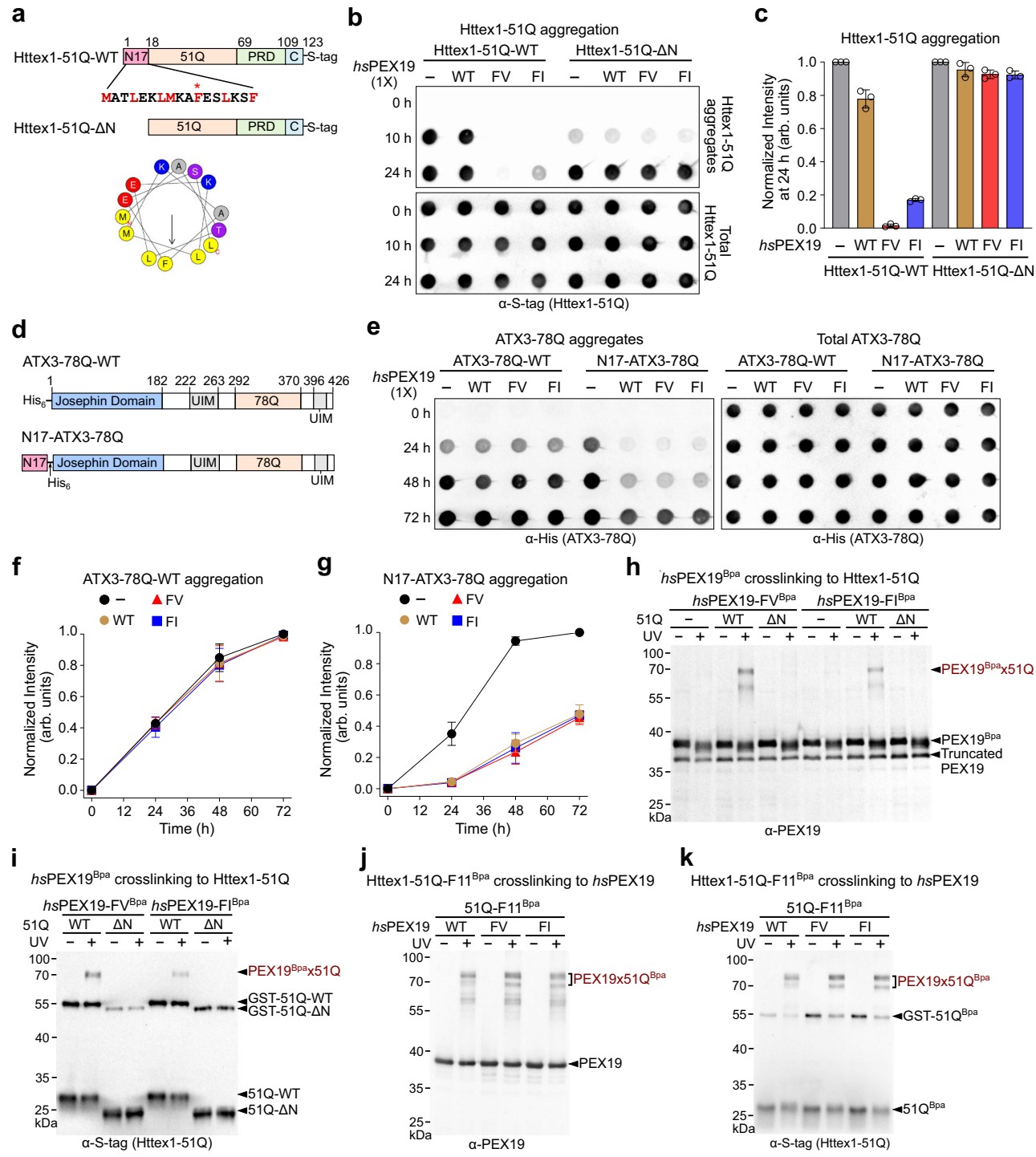

**Fig. 3 | The α4 helix of *hs*PEX19 variants directly interacts with the N17 domain of mHttex1. a** N-terminal GST-tagged Httex1-51Q-WT or Httex1-51Q-ΔN were used to monitor Htt51Q aggregation or interaction between *hs*PEX19 and Httex1-51Q proteins in vitro. The helical wheel illustrates the distribution of hydrophobic amino acids in the amphipathic helix of the N17 domain of Httex1-51Q. The sequences of hydrophobic amino acids in the N17 domain are also highlighted in red. "*" represents the Bpa incorporation site on Httex1-51Q. **b, c** In vitro aggregation assay of Httex1-51Q-ΔN in the absence and presence of *hs*PEX19 variants. SDS-insoluble aggregates of Httex1-51Q in (**b**) and their replicates were quantified and shown in (**c**) (*n* = 3, mean ± SD). **d** Schematic illustration of Ataxin3-78Q-WT and N17-Ataxin3-78Q. Ataxin3 consists of an N-terminal Josephin domain, several ubiquitin-interacting motifs (UIMs), and a polyQ repeat domain. The N17 domain of Httex1 was fused to the N-terminus of Ataxin3-78Q. **e–g** In vitro aggregation assay of Ataxin3-78Q-WT and N17-Ataxin3-78Q in the absence and presence of *hs*PEX19

variants. 30 μM of Ataxin3-78Q was incubated with 30 μM of *hs*PEX19 proteins at 37 °C. SDS-insoluble aggregates of Ataxin3-78Q-WT and N17-Ataxin3-78Q in (**e**) and their replicates were quantified and shown in (**f**) and (**g**), respectively (*n* = 3, mean ± SD). **h, i** Bpa crosslinking assay to monitor the direct association of *hs*PEX19[Bpa] with Httex1-51Q-WT or Httex1-51Q-ΔN. 3 μM of *hs*PEX19-FV[Bpa] or *hs*PEX19-FI[Bpa] was incubated with an equimolar concentration of Httex1-51Q-WT or Httex1-51Q-ΔN for 3 h at 30 °C. Crosslinked samples were analyzed using Western blots probed with PEX19 (**h**) and S-tag (51Q) (**i**) antibodies. **j, k** Bpa crosslinking assay to monitor the intermolecular interaction of Httex1-51Q-F11[Bpa] with *hs*PEX19-WT and its *hs*PEX19 variants. 3 μM of Httex1-51Q-F11[Bpa] was incubated with 1.5 μM of *hs*PEX19 proteins for 3 h at 30 °C. Crosslinked samples were subjected to Western blot analysis against PEX19 (**j**) and S-tag (51Q) (**k**) antibodies. All Bpa crosslinking assays in (**h–k**) were performed twice or three times independently (*n* = 3 for (**h** and **i**) or 2 for (**j** and **k**)). Source data are provided as a Source Data file.

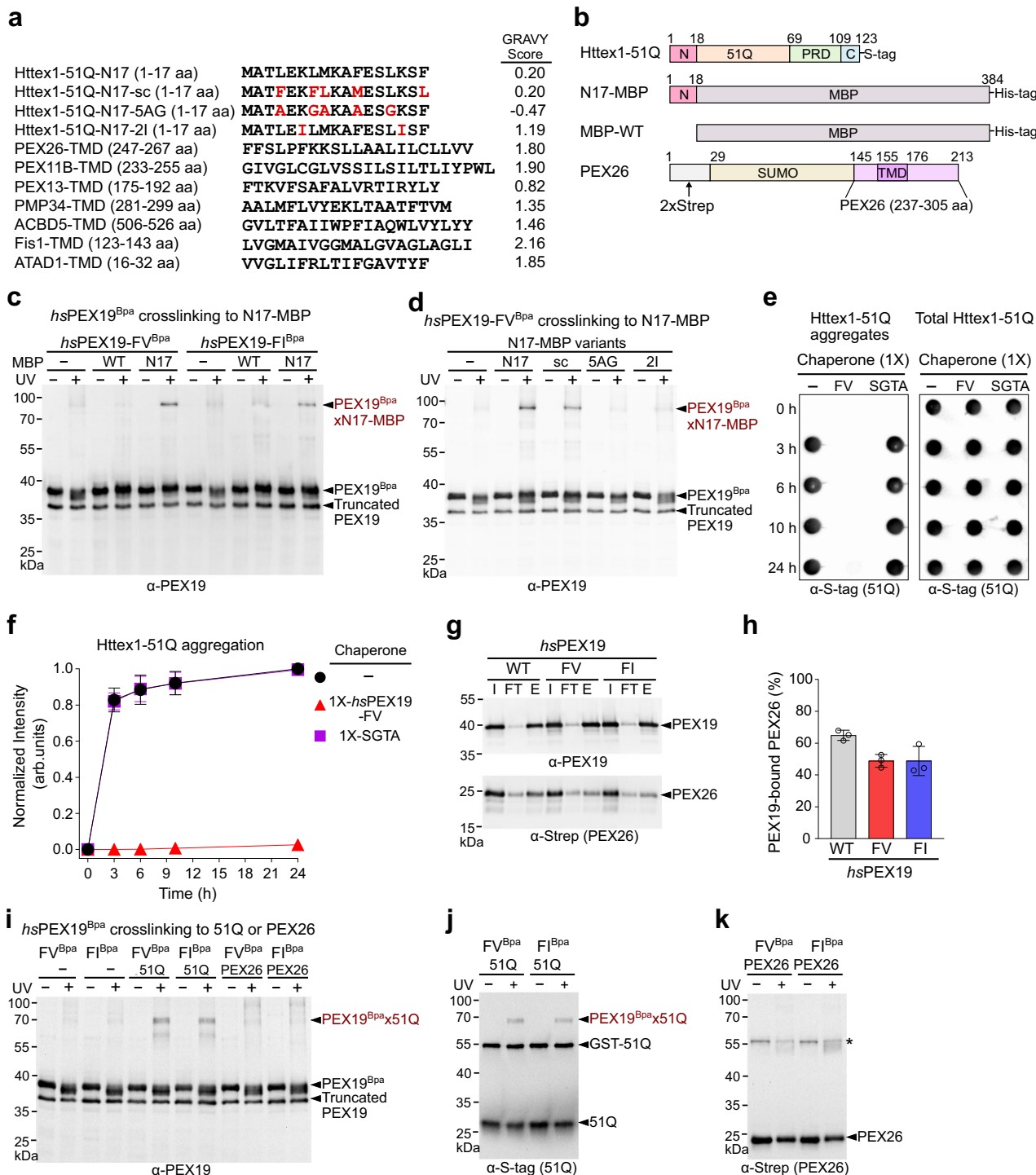

**Fig. 4 | The α4 helix of *hs*PEX19 variants is a specific binding site for the N17 domain of Httex1-51Q. a** Sequences of the N17 domain of Httex1-51Q and TMDs of peroxisomal and mitochondrial membrane proteins and their Grand Average of Hydropathy (GRAVY) scores[103]. All listed membrane proteins are known to interact with *hs*PEX19 during their targeting to peroxisome or mitochondria[39,46,58,59]. PEX26, PEX11B, and ACBD5 are peroxisomal tail-anchored membrane proteins (TAs), whereas Fis1 is a dually localized TA in mitochondria and peroxisomes[58,59]. PEX13 and PMP34 are multi-spanning peroxisomal membrane proteins (PMPs)[39,59]. ATAD1 is an N-terminal signal-anchored membrane protein localized in both mitochondria and peroxisome[46]. **b** Schematic representation of Httex1-51Q, N17-MBP, MBP-WT, and PEX26. The isolated N17 sequence was N-terminally fused to MBP (maltose binding protein). The recombinant PEX26 protein contains the N-terminal 2 × Strep-tagged SUMO domain and the PEX26 targeting sequences (237–305 aa) encompassing the TMD and C-terminal charged tail of PEX26[63]. **c**, **d** Bpa

crosslinking assay of *hs*PEX19[Bpa] with MBP-WT, N17-MBP, N17-MBP variants. **e**, **f** In vitro aggregation assay to monitor the chaperone activity of *hs*PEX19-FV and SGTA toward Httex1-51Q. SDS-insoluble Httex1-51Q aggregates in (**e**) and their replicates were quantified and shown in (**f**) (*n* = 3, mean ± SD). **g**, **h** Representative images of His₆-PEX19 pulldown assay with PEX26 in (**g**) and their quantification in (**h**). I, FT, and E denote total input, flowthrough, and elution, respectively. The amounts of PEX26 bound to *hs*PEX19 were analyzed by western blot against Strep (PEX26) and PEX19 antibodies. Data in (**h**) are shown as mean ± SD, with *n* = 3 (technical replicates). **i**–**k** Bpa crosslinking assay of *hs*PEX19[Bpa] with either Httex1-51Q or PEX26. The Bpa crosslinking assays with Httex1-51Q were carried out under the same conditions as Fig. 3h and i. Prior to the Bpa crosslinking assay, 0.75 μM of PEX26 was incubated with 3 μM of *hs*PEX19[Bpa] at room temperature for 5 min. "*" represents the SDS-resistant PEX26 dimers in (**k**). All Bpa crosslinking assays were performed three times independently (*n* = 3). Source data are provided as a Source Data file.

whereas substituting two isoleucine residues for the lysine residues of N17 domain (N17-2I-MBP) generates a moderately hydrophobic PMP-TMD-like N17 domain (Fig. 4a). Among the three N17-MBP mutants, only N17-sc-MBP displayed a crosslinked band with *hs*PEX19-FV[Bpa], suggesting that the hydrophobicity of the N17 domain is crucial for the interaction with the α4 helix of *hs*PEX19 (Fig. 4d). In contrast to *hs*PEX19-FV, SGTA, a cytosolic co-chaperone that binds highly hydrophobic TMDs of ER TAs[60,61], was unable to suppress the aggregation of Httex1-51Q (Fig. 4e, f). Collectively, our results suggest that mutations on the α4 helix of *hs*PEX19 enable binding to the relatively low hydrophobic N17 domain of mHttex1.

Several studies suggested that the α1 helix of PEX19-CTD serves as a binding site of PMPs[41,43,62] (Supplementary Fig. 10a). We also previously showed that the M179 residue in the α1 helix of PEX19-CTD interacts with PEX26, the peroxisomal TA[63] (Supplementary Fig. 10a). Given that Httex1-51Q binds to the α4 helix of *hs*PEX19 variants, we checked whether the *hs*PEX19 variants also interact with a bona fide PEX19 substrate, PEX26[40]. At an approximately 3-fold excess concentration of the endogenous *hs*PEX19[64], the amounts of PEX26 loaded onto *hs*PEX19 variants were comparable to *hs*PEX19-WT, indicating that these mutations on the α4 helix of *hs*PEX19 do not largely alter the overall binding capacity of PEX26 (Fig. 4g, h). Consistent with the results of the His-PEX19 pulldown assay, the *hs*PEX19-M179[Bpa]-FV and -FI variants readily crosslinked with PEX26 (Supplementary Fig. 10b–d). In contrast to *hs*PEX19-M179[Bpa] variants, both *hs*PEX19-FV[Bpa] and *hs*PEX19-FI[Bpa] did not crosslink to PEX26 (Fig. 4i, k), suggesting that the α1 helix of PEX19 is a primary binding site for PEX26. Conversely, in contrast to *hs*PEX19-FV[Bpa] and *hs*PEX19-FI[Bpa] (Fig. 4i, j), *hs*PEX19-M179[Bpa] variants did not show a distinct crosslinked band with Httex1-51Q (Supplementary Fig. 10e, f). Taken together, these results suggest that F255 and V259/I259 mutations on *hs*PEX19 can create a specific binding site for the N17 domain of Httex1-51Q, eventually resulting in robust suppression activity for Httex1-51Q aggregation.

We tested whether *hs*PEX19-FV also prevents aggregation of a non-polyQ protein, TDP43, which is associated with another neurodegenerative disease, ALS. To this end, we performed an established in vitro aggregation assay using the purified TDP43-TEV-MBP-His₆ protein[65]. The addition of TEV protease enables the initiation of TDP43 aggregation (Supplementary Fig. 11a, black). In contrast to Httex1-51Q aggregation in Fig. 2d, incubation with *hs*PEX19-WT or *hs*PEX19-FV exhibited only a minor delay in TDP43 aggregation kinetics (Supplementary Fig. 11a, blue and red). To further monitor TDP43 aggregation in cells, we generated a stable HEK293 cell line (TDP43-BiFC) that expresses both TDP43-VN and TDP43-VC. Given that phosphorylation and acetylation on TDP43 promote its aggregation[66–69], we used Forskolin as a phosphorylation activator and Apicidin as an acetylation-inducing agent for TDP43[70,71]. Treatment with either Forskolin or Apicidin significantly increased the fluorescence intensities of TDP43-BiFC in the cytosol (Supplementary Fig. 11b, c). Overexpression of *hs*PEX19-WT or *hs*PEX19-FV showed at most a minor rescue of Forskolin or Apicidin-induced cytosolic TDP43 aggregation in HEK293 cells (Supplementary Fig. 11d–g). Together with Fig. 2, we conclude that *hs*PEX19-FV selectively suppresses the aggregation of mHttex1 in vitro and in mammalian cells.

## *hs*PEX19-FV rescues HD-associated phenotypes

Since the in vitro data in Fig. 2 indicated that *hs*PEX19-FV has a higher chaperone activity for mHttex1 than *hs*PEX19-FI, we next assessed whether *hs*PEX19-FV can rescue the proteotoxicity of mHttex1 in primary neurons and *Drosophila* HD models. To test whether *hs*PEX19-FV protects striatal neurons from mHttex1 proteotoxicity, we coexpressed Httex1-134Q-GFP with *hs*PEX19-WT or *hs*PEX19-FV at 7 days in vitro (DIV) in primary striatal neurons (Fig. 5a). In contrast to Httex1-19Q-GFP- and vector control-coexpressing striatal neurons,

at 48 h post-transfection, we observed largely fragmented neurites in the striatal neurons when coexpressed with Httex1-134Q-GFP and vector control, suggesting that mHttex1 induces neuritic degeneration[72,73] (Fig. 5a, b). Over 80% of striatal neurons coexpressing Httex1-134Q-GFP and *hs*PEX19-FV exhibited unfragmented healthy neurites, while various degrees of fragmented neurites were observed in the Httex1-134Q-GFP-and *hs*PEX19-WT-coexpressing neurons (Fig. 5a, b). Furthermore, the results of the TUNEL cell death assay showed that *hs*PEX19-FV significantly reduced apoptotic DNA fragmentation in Httex1-134Q-GFP expressing neurons (Supplementary Fig. 12a, b). Therefore, these results suggest that *hs*PEX19-FV effectively protects against mHttex1-induced neurotoxicity in mouse striatal neurons.

We next tested whether the *hs*PEX19-FV variant could rescue HD-associated phenotypes in *Drosophila* HD models. To this end, we generated transgenic fly lines expressing pACU2 empty vector (vector control), *hs*PEX19-WT, or *hs*PEX19-FV and coexpressed Httex1-20Q or Httex1-93Q under the control of *Elav-GAL4* (pan-neurons) or *D42-GAL4* (motor neurons) drivers (Supplementary Table 1). As a negative control, we used the *W[1118]* fly line which does not carry a Httex1 transgene. Compared to *W[1118]/vector control* and *Httex1-20Q/vector control* flies, motor- or pan-neuronal Httex1-93Q overexpression in *Httex1-93Q/vector control* flies led to a significant defect in their locomotion capacities (Fig. 5c, d). In contrast to *hs*PEX19-WT, *hs*PEX19-FV expression partially restored the impaired climbing ability of flies overexpressing Httex1-93Q in motor- and pan-neurons (Fig. 5c, d). Overexpression of *hs*PEX19-FV marginally increased the lifespan of *W[1118]* flies only at later time points despite no significant effect observed in *Httex1-20Q* flies (Fig. 5e, f). In contrast, *hs*PEX19-FV significantly increased the lifespan of flies expressing Httex1-93Q (Fig. 5g). *hs*PEX19-WT exhibited a modest increase in survival rates in both *Httex1-20Q* and *Httex1-93Q* flies (Fig. 5f, g). Overall, these results indicated that *hs*PEX19-FV rescued HD-relevant behavioral deficits and improved the survival of HD flies.

We further addressed the question of whether *hs*PEX19-FV has a protective role in the mHttex1 aggregation-associated neurotoxicity in HD flies. The results of immunohistochemistry showed that the numbers of Httex1-93Q-positive puncta in *Httex1-93Q/hsPEX19-FV* flies were significantly reduced in both motor- and pan-neurons compared to *Httex1-93Q/vector control* flies (Supplementary Fig. 13a–h). Furthermore, despite the exclusive cytosolic localization of *hs*PEX19-FV, *Httex1-93Q/hsPEX19-FV* flies displayed nuclear-localized soluble Httex1-93Q in both motor- and pan-neurons (Supplementary Fig. 13g, h). Consistent with these results, overexpression of *hs*PEX19-FV in *Httex1-93Q* flies at 15 days of age revealed a notable enhancement in their locomotion and a significant decrease in mHttex1 aggregation levels in the HD fly brains (Supplementary Fig. 14a–c). In contrast, *hs*PEX19-WT exhibited no significant differences in Httex1-93Q aggregation despite a mild improvement in the climbing ability of HD flies (Supplementary Fig. 14a–c). More importantly, *hs*PEX19-FV ameliorated mHttex1 aggregation-associated neurotoxicity more effectively than *hs*DNAJB1, a known mHttex1 aggregation suppressor[74,75] (Supplementary Fig. 15a–c). Taken together, *hs*PEX19-FV effectively reduced mHttex1 aggregation-induced neurotoxicity, thereby leading to neuroprotection in mouse striatal neurons and improved survival and locomotion in HD flies.

## Discussion

Here, we used the yeast toxicity-based screening method to identify two yeast PEX19 variants, *sc*PEX19-FV (L288F/E292V) and *sc*PEX19-FI (L288F/E292I), that rescue the toxicity of mHttex1 in yeast. Since the sites of these mutations in the α4 helix of PEX19 are highly conserved, we further generated the human variants *hs*PEX19-FV (M255F/Q259V) and *hs*PEX19-FI (M255F/Q259I). We confirmed that *hs*PEX19 variants effectively suppress mHttex1 aggregation in vitro and in mammalian cells. The mutated phenylalanine residue in the α4 helix of *hs*PEX19

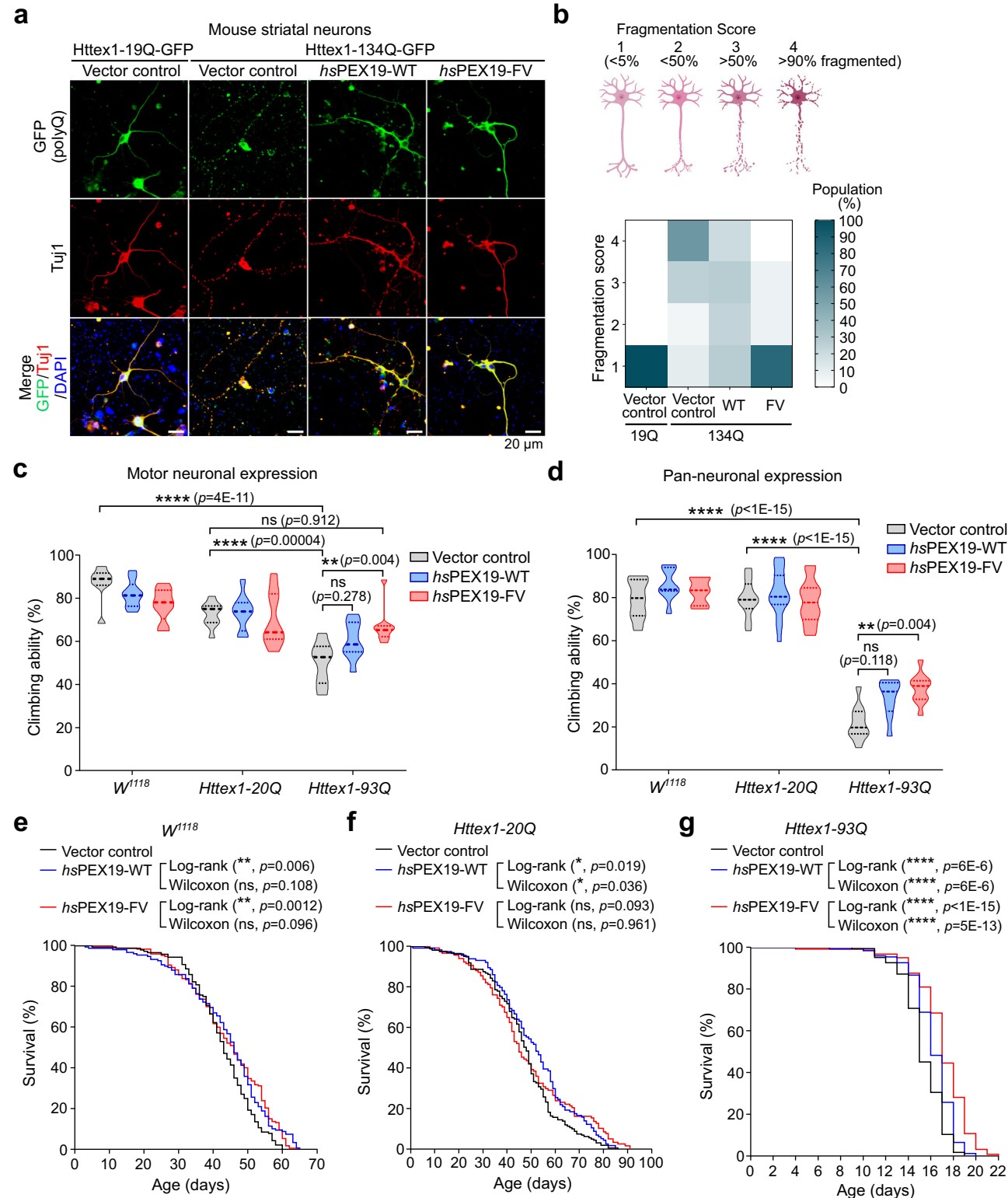

variants directly interacts with the N17 domain of mHttex1, thereby significantly delaying the aggregation of mHttex1. Finally, our results demonstrate that *hs*PEX19-FV rescues mHttex1-induced neuritic degeneration in primary striatal neurons and HD-associated behavioral deficits and lifespan in the *Drosophila* HD models.

Several chaperones have been identified as mHttex1 aggregation suppressors, which target different domains of mHttex1. Previous studies showed that the TRiC chaperonin and Hsc70 chaperone bind to the N17 domain of mHttex1, thereby preventing mHttex1

aggregation[16,76]. In addition, two J-domain proteins (JDPs), DNAJB6 and DNAJB8, and the β subunit of the nascent polypeptide-associated complex (NAC) directly interact with the PolyQ repeat domain, thereby suppressing polyQ-mediated aggregation[49,77–79]. Furthermore, a recent study showed that another JDP, DNAJB1, together with Hsc70 and Apg2, binds to the PRD of mHttex1, and the trimeric chaperone system prevents and redissolves mHttex1 aggregates[74]. The S/T-rich region of DNAJB6 was suggested to form hydrogen bonds with the polyQ residues[77], whereas the positively charged N terminus of βNAC is

**Fig. 5 | *hs*PEX19-FV mitigates mHttex1-induced neurodegenerative phenotypes. a** Confocal microscopy images of mouse striatal neurons coexpressing Httex1-19Q-GFP and Httex1-134Q-GFP with vector control, *hs*PEX19-WT, or *hs*PEX19-FV. Tuj1 (neuron marker) was stained using Tuj1 antibody. Scale bar: 20 μm. **b** The degrees of fragmentation in a primary neuron are classified as four fragmentation scores: score 1 (fragmented areas in a single neuron are less than 5%), score 2 (5% < fragmented areas < 50%), score 3 (50% < fragmented areas < 90%), score 4 (fragmented areas > 90%). The heatmap shows the population of fragmentation scores for each condition (from left to right, $n = 23, 19, 25, 27$ neurons). **c** Climbing ability of 12-day-old adult flies ($W^{1118}$, *Httex1-20Q*, and *Httex1-93Q*) expressing vector control, *hs*PEX19-WT, and *hs*PEX19-FV in motor neurons. The data in (**c**) are shown as violin plots with mean and quartiles (from left to right, $n = 102, 120, 119, 105, 103, 110, 101, 112, 105$ adult flies). Climbing index (5 cm/5 sec). Statistical significance was evaluated using the two-way ANOVA with Tukey post-hoc test. **$p < 0.01$,

$****p < 0.0001$, ns = not significant. **d** Climbing ability of 10-day-old adult flies ($W^{1118}$, *Httex1-20Q*, and *Httex1-93Q*) expressing vector control, *hs*PEX19-WT, and *hs*PEX19-FV in pan-neurons. Climbing index (5 cm/5 sec). The data in (**d**) are shown as violin plots with mean and quartiles (from left to right, $n = 109, 102, 91, 251, 205, 152, 125, 103, 130$ adult flies). Statistical significance was evaluated using two-way ANOVA with Tukey post-hoc test. **$p < 0.01$, ****$p < 0.0001$, ns = not significant. **e–g** Lifespan analysis of $W^{1118}$, *Httex1-20Q*, and *Httex1-93Q Drosophila* expressing vector control, *hs*PEX19-WT, and *hs*PEX19-FV in pan-neurons. Lifespan data were plotted as Kaplan-Meier survival curves, and *p*-values were determined using the Log-rank (Mantel-Cox) test and Gehan-Breslow-Wilcoxon test. Vector control, *hs*PEX19-WT, *hs*PEX19-FV; (**e**) $n = 140, 148, 167$, (**f**) $n = 159, 114, 117$, (**g**) $n = 164, 151, 121$ adult flies. *$p < 0.05$, **$p < 0.01$, ****$p < 0.0001$, ns = not significant. Genotypes of *Drosophila* used in (**c–g**) are listed in Supplementary Table 1. Source data are provided as a Source Data file.

---

involved in interactions with the polyQ repeat domain[49]. These polyQ-binding domains in DNAJB6 and βNAC are low complexity linkers located between the JD and C-terminal substrate binding domain and N-terminal unstructured small domain (~ 40 aa), respectively[49,77,80]. In this study, we showed that the hydrophobic interactions between the α4 helix of *hs*PEX19 variants and the N17 domain of mHttex1 enable the suppression of mHttex1 aggregation. In contrast to DNAJB6 and βNAC that form electrostatic interactions or hydrogen bonds with mHttex1, the amphipathic N17 domain appears to dock into the hydrophobic farnesyl group-binding groove of *hs*PEX19-CTD[43]. The hydrophobic residues in the N17 domain are likely to be protected from aqueous cytosolic environments, thereby inhibiting the self-assembly of mHttex1. Together with these previous studies, our results further suggest that, depending on their mHttex1-binding domains, chaperones can employ different molecular mechanisms to prevent mHttex1 aggregation.

Membrane protein chaperones recognize their cargo membrane proteins primarily based on the hydrophobicity and location of TMDs[81,82]. TMDs are typically between 15 and 30 amino acids long with widely variable hydrophobicity. Regardless of TMD location, PEX19 generally recognizes moderately hydrophobic TMDs in peroxisomal and mitochondrial membrane proteins[39,41,46,58,59] (Fig. 4a). In contrast, SGTA (Sgt2 in yeast) and TRC40 (Get3 in yeast) preferentially bind to more hydrophobic TMDs located near the C-terminus in the ER TAs[41,60,82–84]. Consistent with the low hydrophobicity and N-terminal localization of the N17 domain of mHttex1 (Fig. 4a), *hs*PEX19-WT exhibits a mild chaperone activity toward Httex1-51Q (Fig. 2d), whereas SGTA was not able to suppress the aggregation of Httex1-51Q (Fig. 4f). The *hs*PEX19 variants suppressed the formation of both SDS-insoluble larger aggregates and fibrils more efficiently than *hs*PEX19-WT (Fig. 2c–f). Despite the unaltered overall binding capacity of *hs*PEX19 variants to PEX26, a hydrophobic residue, F255, in the α4 helix of *hs*PEX19 variants did not interact with PEX26 (Fig. 4g–k). In contrast, the α1 helix of *hs*PEX19, the primary binding site of PMPs[41,43,62], did not bind to the N17 domain (Supplementary Fig. 10e, f), suggesting that the α4 helix of *hs*PEX19 variants is a specific binding site for the N17 domain of mHttex1. Alternatively, the α4 helix of *hs*PEX19 could be a unique structural feature that discriminates membrane proteins with low hydrophobicity of TMDs. Nevertheless, further structural analysis of *hs*PEX19 variants would explain how *hs*PEX19 variants efficiently suppress mHttex1 aggregation.

Accumulation of mHtt aggregates in the cytoplasm sequesters a variety of cytosolic proteins, thereby interfering with diverse cellular functions and endomembrane structures[20,21,23,85,86]. Several studies showed that cytoplasmic mHtt aggregates impair nucleocytoplasmic transport of proteins and mRNAs by sequestering nuclear-shutting factors to the aggregates[85,86]. Furthermore, cytoplasmic mHtt aggregates trap with the cytoskeletal transport system as well as other polyQ proteins in the cytosol, thus further disrupting axonal transport in a *Drosophila* HD model[87]. Despite the predominant nuclear localization

of mHttex1 aggregates in Httex1-93Q expressing *Drosophila*, our results showed that overexpressing the *hs*PEX19-FV variant in the cytosol significantly reduces the nuclear aggregation of mHttex1 in both motor- and pan-neurons (Supplementary Fig. 13). Prior to the nuclear import of mHttex1, *hs*PEX19-FV could prevent the aggregation of mHttex1 in the cytosol (Supplementary Fig. 7e, f). Together with previous studies[85–87], our results suggest that maintaining a soluble form of mHttex1 assisted by molecular chaperones in the cytosol could also modulate the conformational quality of nuclear mHttex1.

Mitigating any potential off-target effects caused by the artificial mutations on a target chaperone would be critical for further therapeutic applications. The identified *hs*PEX19 variants appear to be specific to HD, relative to proteins linked to other neurodegenerative diseases, potentially due to both the highly conserved sequences[88] and the amphipathic helix property of N17 domain of Httex1. *hs*PEX19-WT and its variants were unable to suppress the aggregation of Ataxin3-78Q (Fig. 3e, f). In addition, *hs*PEX19-FV displayed only a very modest chaperone activity in TDP43 in vitro aggregation assays (Supplementary Fig. 11a). Substituting the E292 residue into the α4 helix of *sc*PEX19 variants with various hydrophobic amino acid residues led to different capacities for ameliorating mHttex1-induced toxicity in yeast (Fig. 1c). These results further suggest that direct modulation of the amino acid sequences on the N17 domain-binding site of PEX19 variants could generate higher substrate specificity for HD. Therefore, further tuning other amino acid sequences in the α4 helix of *hs*PEX19-FV would help to eliminate unidentified side effects caused by *hs*PEX19 variants for HD.

The rescuing effects of *hs*PEX19-FV observed in mHttex1-expressing flies might not be entirely due to the increased chaperone action of *hs*PEX19-FV on the N17-mediated mHttex1 aggregates. We note that the PEX26 binding capacity of *hs*PEX19-FV was not drastically altered compared to *hs*PEX19-WT (Fig. 4g, h), suggesting that the variant can act on peroxisomal membrane proteins as well as mHttex1. Indeed, given that a variety of cytosolic chaperones are sequestered to mHttex1 aggregates[20,21], as also observed with *hs*PEX19-WT in Httex1-134Q expressing HEK293T cells (Supplementary Fig. 7f), overexpression of *hs*PEX19-FV could assist PMP targeting to the peroxisome, thereby maintaining proper peroxisome biogenesis in mHttex1-expressing flies.

Overall, our study demonstrates that engineering an ATP-independent membrane chaperone is a feasible approach to reducing N17-mediated mHttex1 aggregates in HD. Several tetratricopeptide repeat (TPR) domain-containing chaperones, i.e., ATP-independent chaperones involved in ER and mitochondrial membrane protein biogenesis, are known to have decreased expression levels in HD patients[7]. Furthermore, a recent study identified the TPR domain-containing chaperones, TTC1 and TOMM70A, as mitochondrial membrane protein biogenesis factors[89]. Thus, our approach could be readily applicable for designing other ATP-independent membrane protein chaperones that specifically reduce mHttex1 toxicity and maintain proper membrane

protein biogenesis for other organelles. Furthermore, given that amphipathic helix-mediated aggregation was previously observed in α-Synuclein, islet amyloid polypeptide, and apolipoprotein C-II[90–93], this approach may not be limited to HD but could also be used for other diseases such as Parkinson's disease, diabetes, and cardiac amyloidosis. Therefore, ATP-independent membrane protein chaperones could serve as a design platform for therapeutic development that targets various diseases associated with misfolding and aggregation of amphipathic helix.

## Methods

### Ethical statement

All experiments involving animal procedures were approved by the Institutional Animal Care and Use Committees (IACUC) of IBS (Daejeon, Korea). No ethical approval is required for experiments with *Drosophila melanogaster*.

### Plasmids

To generate yeast integration plasmids for pRS306-Gal-FLAG-Httex1-25QΔP-GFP and pRS306-Gal-FLAG-Httex1-97QΔP-GFP, the insert genes for FLAG-Httex1-25QΔP-GFP and FLAG-Httex1-25QΔP-GFP were amplified from pYES2-FLAG-Httex1-25QΔP-GFP and pYES2-FLAG-Httex1-97QΔP-GFP (gift from F. Ulrich Hartl Lab)[19], respectively. The vector backbone for the pRS306-Gal gene was amplified from pRS306-Gal-NDC80-RFP-MPS1 (gift from Won-ki Huh Lab)[94]. Gibson assembly was performed to incorporate the insert genes into the pRS306-Gal vector. For pRS413-Gal-*sc*PEX19, the *sc*PEX19 gene was amplified from the isolated yeast genomic DNA and further incorporated into pRS413-Gal vector using Gibson assembly. For His$_6$-*sc*PEX19 and His$_6$-*hs*PEX19, the yeast PEX19 gene and the human PEX19 gene were cloned into pET-33b, respectively.

### Yeast strains

Yeast strain used in this study is *W303a* (MATa, can1-100, his3-11,15, leu2-3,112, trp1-1, ura3-1, ade2-1; gift from Won-ki Huh Lab). To generate FLAG-Httex1-25QΔP-GFP and FLAG-Httex1-97QΔP-GFP integrated-*W303a* strains, the linearized pRS306-Gal-FLAG-Httex1-25QΔP-GFP and pRS306-Gal-FLAG-Httex1-97QΔP-GFP by NcoI were transformed into *W303a* strain and selected on SD media lacking Ura.

### Spotting assay

Yeast cells were grown to mid-log phase in the selective media containing 2% raffinose and 0.1% glucose. Overnight cultured cells were diluted to OD$_{600}$ of 0.1. The diluted cells were spotted onto 3% galactose and 1% raffinose-containing plates in serial 5-fold dilutions. Equal spotting was confirmed by spotting the same diluted cells on plates containing 2% glucose. After 2-3 days of incubation at 30 °C, the images were acquired using the iBright™ FL 1000 imaging system (Thermo Fisher Scientific).

### Yeast PEX19 library generation

A random mutagenesis library was generated by error-prone PCR using a GeneMorph II mutagenesis kit (Agilent, #200552). The cloned wild-type *sc*PEX19 gene was used as a template, and the reactions were performed according to the manufacturer's protocol. To generate diverse mutations, two different amounts of template (5 ng and 50 ng) were used in PCR reactions for 34 cycles, followed by two more sequential PCR reactions. The resulting PCR products in each reaction were re-amplified using Q5 High-Fidelity DNA Polymerase (New England Biolabs, #M0491L) and gel-purified using a QIAquick Gel Extraction Kit (Qiagen, #28704). The gel-purified PCR products were inserted into the pRS413 vector using Gibson assembly. The Gibson assembly mixture was desalted and then transformed into DH5α competent cells using electroporation. The transformed colonies were pooled and purified using a Midi prep kit (MACHEREY-NAGEL, # 740410.10).

### Yeast toxicity-based screening

The *sc*PEX19 plasmid library was transformed into Httex1-97QΔP strain using the LiAc transformation method[95], and the cells were spread onto a 2% glucose-containing SD-His-Ura plate. Approximately 90,000 colonies were pooled and further cultured in SR-His-Ura media supplemented with 2% raffinose and 0.1% glucose overnight at 30 °C. The culture cells were adjusted OD$_{600}$ to 0.004 and spread onto an SG-His-Ura plate (3% galactose and 1% raffinose). We used the *sc*PEX19-WT-transformed Httex1-97QΔP and Httex1-25QΔP cells as negative and positive controls, respectively. The selected colonies from the SG-His-Ura plate were further confirmed using the spotting assay. The *sc*PEX19 variants were amplified using colony PCR, and then mutation sites were identified with sequencing. The selective *sc*PEX19 variants were generated by Quick Change mutagenesis using pRS413-Gal-*sc*PEX19-WT as a template. Those plasmids were transformed into Httex1-97QΔP and Httex1-25QΔP cells and further confirmed their suppression activity using the spotting assay.

### Preparation of yeast cell extracts

For total yeast cell extract, the cell pellets were resuspended in 100 μL of 0.3 M NaOH and incubated for 3 min at room temperature. After washing the cells with water, the cell pellets were resuspended in 100 μL of lysis buffer (20 mM Tris (pH 8.0), 150 mM NaCl, 2% SDS) supplemented with protease inhibitor cocktail (cOmplete mini, EDTA-free protease inhibitor cocktail, Roche, # 11836170001) and then incubated at 95 °C for 5 min. After centrifugation at 15,000 × *g* for 5 min, the clarified lysate was subjected to Western blot analysis.

To monitor SDS-insoluble 97Q aggregates in yeast, the cell extracts were prepared as described with minor modifications[96]. The cells were induced at OD$_{600}$ of 0.1 with 1% galactose and 3% raffinose for 4 h. The cell pellets were resuspended in 200 μL of lysis buffer (25 mM Tris (pH 8.0), 150 mM NaCl, 1 mM EDTA, 0.5% Triton X-100, 5% glycerol) supplemented with protease inhibitor cocktail and benzonase, and then lysed by Distruptor Genie (Scientific industries) with glass beads. The extracts were clarified by centrifugation of 500 × *g* for 5 min at 4 °C.

### Protein expression and purification

Expression and purification of GST-Httex1-51Q-WT and GST-Httex1-51Q-ΔN proteins were performed as described[18]. GST-Httex1-51Q-WT and GST-Httex1-51Q-ΔN were expressed BL21 Star™ (DE3) (Invitrogen, #C601003) with 1 mM IPTG at 18 °C for 2.5 h. Cells were resuspended in PBS supplemented with 150 mM NaCl, 1 mM EDTA, and cOmplete™ EDTA-free protease inhibitor cocktail (Roche, #11836170001). After sonication, the lysate was centrifuged at 20,000 × *g* for 20 min at 4 °C. The supernatant was incubated with glutathione agarose resin (Thermo Scientific, #16100) for 1 h at 4 °C. The resin was washed with PBS containing 500 mM NaCl, 5 mM MgCl$_2$, 2 mM ATP, and then the protein was eluted with 15 mM glutathione dissolved in PBS. After dialysis in the buffer (50 mM Tris-HCl (pH 8.0), 100 mM NaCl, 5% glycerol), the purified GST-Httex1-51Q proteins were concentrated with Amicon® Ultra 30,000 MWCO centrifugal filters and 0.22 μm-filtered through prior to storing at − 80 °C.

His$_6$-*hs*PEX19 or His$_6$-*sc*PEX19 proteins were expressed in BL21 Star™ (DE3) with 0.5 mM IPTG at 18 °C overnight. His$_6$-SGTA, MBP-His$_6$, and N17-MBP-His$_6$ proteins were expressed in BL21 Star™ (DE3) with 0.1 mM IPTG for 3 h at 37 °C. Cells were resuspended in Buffer A (20 mM Tris-HCl (pH 8.0), 300 mM NaCl, 5 mM imidazole, 2 mM 2-mercaptoethanol, 10% glycerol) supplemented with cOmplete™ EDTA-free protease inhibitor cocktail and lysed using sonication. The clarified lysate was incubated with Ni-NTA agarose resin (Qiagen, #3023), and the proteins were eluted with 300 mM imidazole dissolved in Buffer A. After dialysis in Buffer B (20 mM HEPES (pH 7.5), 150 mM NaCl, 2 mM 2-mercaptoethanol, 10% glycerol), the purified proteins were stored at − 80 °C.

To site-specifically incorporate $p$-benzoyl-L-phenylalanine (Bpa) into His$_6$-$hs$PEX19 or GST-Httex1-51Q proteins, the coding sequences for the residue F255 in the His$_6$-$hs$PEX19 variants or the residue F11 in GST-Httex1-51Q were replaced with an amber codon (TAG) using npfu-special polymerase (Enzynomics, #P100S) according to the manufacturer's introduction. Expression plasmids for His$_6$-$hs$PEX19-FV$^{amb}$ or His$_6$-$hs$PEX19-FI$^{amb}$, or GST-Httex1-51Q-F11$^{amb}$ and tRNA$_{CUA}^{Opt}$ synthetase[57] were co-transformed into BL21 Star$^{TM}$ (DE3) cells. The expression of tRNA synthetase was induced with 0.2% arabinose at $OD_{600}$ of 0.3. At $OD_{600}$ of 0.6, proteins were induced with 0.5 mM IPTG and 1 mM Bpa (Bachem, #4017646) at 18 °C overnight. Bpa incorporation into the proteins was confirmed by SDS-PAGE analysis. His$_6$-$hs$PEX19-FV$^{Bpa}$, His$_6$-$hs$PEX19-FI$^{Bpa}$, and GST-Httex1-51Q-F11$^{Bpa}$ were purified in the same way as their non-Bpa proteins.

His$_6$-Ataxin3-78Q (gift from Sheena Radford lab) and N17-His$_6$-Ataxin3-78Q were expressed in BL21 Star$^{TM}$ (DE3) with 0.5 mM IPTG at 30 °C for 3 h. His$_6$-Ataxin3-78Q and N17-His6-Ataxin3-78Q were purified using Ni-NTA, the same procedure used for His$_6$-PEX19 proteins. The eluted proteins were loaded onto a superdex$^{TM}$ 200 increase 10/300 column (Cytiva, #28990944), and the monomer fractions were collected and further concentrated with Amicon® Ultra 50,000 MWCO centrifugal filter. The purified Ataxin3-78Q proteins were snap-frozen and used for the filter trap assay.

TDP43-TEV-MBP-His$_6$ (Addgene plasmid #104480) was expressed BL21 Star$^{TM}$ (DE3) with 0.1 mM IPTG overnight at 18 °C. The purification of TDP43-TEV-MBP-His$_6$ was carried out as described[65] with minor modifications. Briefly, the cells were resuspended in Buffer C (20 mM Tris-HCl (pH 8.0), 1 M NaCl, 5 mM imidazole, 2 mM 2-mercaptoethanol, 10% glycerol) and then sonicated. The bound TDP43-TEV-MBP-His$_6$ protein onto Ni-NTA resin was eluted with 300 mM imidazole dissolved in Buffer C. The eluted proteins were loaded onto a superdex$^{TM}$ 200 increase 10/300 column and further purified in Buffer D (20 mM Tris-HCl (pH 8.0), 300 mM NaCl, 1 mM DTT). The purified TDP43-TEV-MBP-His$_6$ protein was concentrated using Amicon® Ultra 50,000 MWCO centrifugal filter and stored at −80 °C.

Expression and purification of 2 × Strep-SUMO-PEX26 (237-305aa) were carried out as described in the previous study[63]. Briefly, the protein was induced with 0.1 mM IPTG in BL21 Star$^{TM}$ (DE3) at 37 °C for 1 h. Cells were lysed by incubating with 0.5% N,N-Dimethyl-1-Dodecanamine-N-Oxide (LDAO, Anatrace, #D360), and 1 × CelLytic$^{TM}$ B Cell Lysis Reagent (Sigma) for 40 min at room temperature. The clarified lysate was then diluted 3-fold with Buffer A and loaded onto a Strep-Tactin Sepharose column (IBA Lifesciences, #2-1201-025). The proteins were eluted with 15 mM $d$-Desthiobiotin (Sigma, #D1411) dissolved in the buffer (20 mM Tris-HCl (pH 8.0), 300 mM NaCl, 2 mM 2-Mercaptoethanol, 0.05% LDAO, 10% glycerol) and further dialyzed in the buffer (20 mM HEPES (pH 7.5), 200 mM NaCl, 10% glycerol, 0.05% LDAO).

The human NAC heterodimer consisting of NACα with an N-terminal 6×His tag and NACβ were expressed and purified as described previously[97], with minor modifications. The cell lysate was loaded onto Ni-NTA resin, and the NAC protein was eluted with 300 mM imidazole. The eluted proteins were further purified using HiTrap Q HP anion exchange chromatography column (Cytiva, #29051325). Elution fractions containing the NAC heterodimer were pooled out and concentrated using an Amicon® Ultra 10,000 MWCO centrifugal filter.

## In vitro aggregation assay – Filter trap assay
3 μM of GST-TEV-Httex1-51Q-Stag proteins were mixed with 1.5 μM or 3 μM of $hs$PEX19 or $sc$PEX19 proteins in the 1X TEV reaction buffer (Invitrogen, #12575015). The polyQ aggregation reaction was initiated by adding 0.05 Units/μL of AcTEV protease (Invitrogen, #12575015) and further incubated at 30 °C. For the in vitro aggregation of Ataxin3-78Q,

30 μM of His$_6$-Ataxin3-78Q were incubated with the equimolar concentration of $hs$PEX19 protein in the reaction buffer (20 mM HEPES (pH 7.5), 25 mM NaCl, 2 mM DTT, 5% glycerol) at 37 °C. Samples were quenched at the indicated time points by mixing an equal volume of the quench buffer (4% (w/v) SDS, 0.1 M DTT) and then boiling at 95 °C for 10 min. The quenched samples were filtered through a 0.22 μm cellulose acetate membrane (Hyundai Micro, #CA020090A) and then washed with 0.1% SDS. The membrane was probed using an S-tag antibody (1:3000 dilution, Invitrogen, #MA1-981) or a His antibody (1:3000 dilution, Genscripts, #A00186) and the secondary mouse antibody IRDye800 (1:15,000 dilution, LiCor, #926-32210). The membrane-trapped polyQ aggregates were detected using the iBright$^{TM}$ FL1000 imaging system.

## In vitro aggregation assay – Turbidity assay
7.5 μM of TDP43-MBP-His$_6$ protein was mixed with either 7.5 μM of $hs$PEX19-WT or $hs$PEX19-FV proteins in the reaction buffer (20 mM HEPES (pH 7.5), 150 mM NaCl, 1 mM DTT). After the addition of AcTEV protease, the optical density values at 395 nm were recorded using a BioTek Epoch 2 plate reader (Agilent).

## ThT (Thioflavin T) fluorescence assay
3 μM of GST-TEV-Httex1-51Q-Stag proteins were mixed with 3 μM of $hs$PEX19 proteins. After supplementing the mixture of 12.5 μM Thioflavin T and 0.05 Units/μL of AcTEV protease, the samples were loaded on a clear, flat bottom 96-well black plate (Corning, #CLS3904). The fluorescence intensity values at an excitation wavelength of 440 nm and emission wavelength of 480 nm were recorded every 15 min using Infinite M Plex microplate (Tecan).

## Negative-stain TEM analysis
3 μM of GST-TEV-Htt51Q-Stag proteins were mixed with 3 μM of $hs$PEX19 proteins in the 1X TEV reaction buffer supplemented with AcTEV protease and incubated at 30 °C for 15 h and 24 h. A copper grid coated with a continuous carbon film (Electron Microscopy Sciences, #CF300-Cu) was negatively glow-discharged at 15 mA for 30 sec. 3 μL of protein samples were applied to a glow-discharged grid, incubated at room temperature for 3 min, and washed twice with distilled water and 0.75% uranyl formate once. The samples were negatively stained with 0.75% uranyl formate for 1 min with gentle shaking. The negatively stained specimens were examined under a FEI Tecnai$^{TM}$ G2 spirit microscope operated at 120 kV. Micrographs were collected using an FEI Eagle 4 K x 4 K CCD camera at a nominal magnification of 15,000X with an electron dose of ~ 30 e-/A2.

## Bpa crosslinking assay
In vitro, aggregation assay with $hs$PEX19-FV$^{Bpa}$/$hs$PEX19-FI$^{Bpa}$ or Httex1-51Q-F11$^{Bpa}$ was performed as described in the section of Filter trap assay. The reaction was stopped by freezing samples at the indicated time points, and frozen reaction aliquots were crosslinked on dry ice ~ 4 cm away from a UVP B-100AP lamp (Analytik Jena) for 10 min. Crosslinked and uncrosslinked $hs$PEX19 or Httex1-51Q proteins were resolved on SDS-PAGE and probed with PEX19 (1:3000 dilution, Novus Biologicals, #NBP2-43757) and S-tag (1:3000 dilution, Invitrogen, #MA1-981) antibodies, respectively.

## Biolayer interferometry (BLI)
The BLI experiments were performed on an Octet R8 Protein Analysis System (Sartorius) at 30 °C. His-$hs$PEX19-FV protein was immobilized using a Ni-NTA biosensor (Sartorius, #18-5101). The biosensor was subsequently incubated in the assay buffer (20 mM HEPES (pH 7.5), 150 mM KoAc) containing various concentrations (ranging from 250 nM to 15.6 nM) of GST-Httex1-51Q proteins for 300 s to allow association. Dissociation was then monitored for 600 s in the assay

buffer. All data were fitted with the 1:1 binding model using the Octet Analysis Studio 12 software (Sartorius) to estimate $k_{on}$, $k_{off}$, and $K_D$ values.

## HEK293 cell culture and plasmid transfection

HEK293T cells (ATCC CRL-3216) were cultured in Dulbecco's Modified Eagle Medium, GlutaMAX™ (DMEM, Gibco, #10569044) supplemented with 10% (v/v) fetal bovine serum (FBS, Gibco, #10082147), 100 U/ml streptomycin, and 100 μ/ml penicillin and incubated in a humidified chamber with 5% $CO_2$ at 37 °C. For coexpression of Httex1-134Q-GFP and hsPEX19 proteins, $3 \times 10^5$ cells per well were seeded on a 6-well plate one day prior to the transfection. Transient transfections of plasmids (each 1.25 μg) were carried out with Polyethylenimine (Polysciences, #23966) according to the manufacturer's instructions.

## Filter trap assay using HEK293T cell lysates

HEK293T cell lysates were prepared for filter trap assay as described previously[98] with minor modifications. Briefly, cells were harvested with PBS supplemented with 1% Triton X-100 and 1x protease inhibitor cocktail (Thermo Scientific, #87785) and further incubated on ice for 30 min. The cell lysates were bath-sonicated in ice water for 5 min. After supplementing with 1% SDS and 50 mM DTT, the cell lysates were heated at 95 °C for 10 min and stored at − 80 °C for filter trap assay and Western blot analysis. Filter trap assays were carried out as described above in the in vitro aggregation assay section. SDS-insoluble Httex1-134Q-GFP aggregates were probed with anti-GFP antibody (1:3000 dilution, Sigma, #SAB4301138) and quantified using iBright Analysis Software (Thermo Scientific). To check protein expression levels, the cell lysates were loaded onto 10% Tris-glycine gels, and then Httex1-134Q-GFP, PEX19, and actin were probed in immunoblots with GFP (1:3000 dilution, Sigma, #SAB4301138), HA (1:3000 dilution, Genscript, #26183), and actin (1:5000 dilution, Invitrogen, #MA5-15452) antibodies, respectively.

## Live cell imaging

HEK293T cells (ATCC CRL-3216) were cultured and cotransfected with Httex1-134Q-GFP and hsPEX19-WT or its variants in a confocal 6-well plate (SPL Life Sciences, #230206). At 48 h post-transfection, the plate was inserted in an inverted Eclipse Ti-E microscope (Nikon) equipped with a stage-top incubator (37 °C, 5% $CO_2$). Live cell images were acquired with a 20× 0.5 NA objective and an automated perfect focus system (PFS).

## Primary neuronal cell culture and transfection

Mice were housed under a 12 h light/dark cycle (light cycle: 7 AM–7 PM) at 20–26 °C with 40–60% humidity. Striata were from mouse pups of ICR mice aged postnatal day 1 (P1). Dissected tissues were digested in an enzyme solution containing 0.1% w/v Papain, 100 μg/mL DNase I, and 1 mM HEPES in Earle's Balanced Salt Solution (EBSS) (Sigma, #E7510-500) at 37 °C for 30 min. After incubation, the enzyme solution was carefully aspirated, and dissected tissues were rinsed with a Neurobasal A medium containing 20% FBS. The tissues were dissociated by mechanical trituration, and the isolated cells were resuspended in the neuro culture medium containing 2% v/v B-27 Supplement, 1 mM L-Glutamine, 1% Penicillin/Streptomycin in Neurobasal A medium (Gibco, #10888-022). $3 \times 10^5$ cells were plated on glass coverslips precoated with 0.1 mg/mL Poly-D-Lysine (Gibco, #A38904-01) and 5 μg/mL Laminin (Gibco, #23017-015) in a 24-well plate. Striatal neurons were cultivated at 37 °C with 5% $CO_2$ in a humidified incubator and used for experiments at 7 days in vitro (DIV). The cells were cotransfected with 1 μg of each plasmid using Lipofectamine™ LTX Reagent with PLUS™ Reagent (Invitrogen, #15338030).

## Immunofluorescence

For HEK293T cells, transfection was carried out under the same conditions for live cell imaging. At 48 h post-transfection, cells were

washed with PBS and fixed in 4% paraformaldehyde (PFA) for 10 min at room temperature. After washing with PBS twice, cells were permeabilized with 0.1% Triton X-100 for 15 min and then blocked with 2% BSA for 1 h at room temperature. Cells were incubated with HA-Tag antibody (1:100 dilution, Invitrogen, #26183) for hsPEX19-WT and its variants proteins and PMP70 antibody (1:200 dilution, Invitrogen, #PA1-650) in 0.1% BSA solution at 4 °C for two overnights. After washing with PBS, the cells were incubated with Alexa Fluor 647 secondary antibody (1:1000 dilution, Invitrogen, #A-21235) for hsPEX19 proteins and Alexa Fluor 568 secondary antibody (1:1000 dilution, Invitrogen, #A-11011) for PMP70 for 1 h at room temperature. To visualize nuclei, cells were additionally stained with 300 nM of DAPI (Invitrogen, #D1306). Images were acquired with an inverted Eclipse Ti-E microscope (Nikon) with a 60× 1.4 NA oil objective.

At 48 h post-transfection, primary neuronal cells were washed with PBS and fixed for 15 min in 4% PFA at room temperature. After washing with PBS twice, the cells were blocked with 10% Normal Donkey Serum (Jackson immunoresearch, #017-000-121) in PBS containing 0.1% Triton X-100 for 1 h at room temperature. Cells were incubated in the blocking solution anti-Tuj1 antibody (1:500 dilution, Abcam, #ab18207) overnight at 4 °C. After washing with PBS, the cells were incubated with Cy3 secondary antibody (1:1000 dilution, Jackson immunoresearch, #706-166-148) for 1 h at room temperature. Nuclei were stained with DAPI.

## TUNEL assay

DNA fragmentation-associated cell death was measured by TUNEL assay[21]. At 48 post-transfection, primary striatal neurons were washed and fixed in 4% PFA for 15 min at room temperature. Neurons were permeabilized in 0.1% Triton X-100 and then further incubated with terminal deoxynucleotide transferase and TMR red dUTP (In Situ Cell Death Detection kit, Roche, #12156792910) for 1 h at 37 °C. Tuj1 and Nuclei were stained using an anti-Tuj1 antibody (1:500 dilution, Abcam, #ab18207) and DAPI, respectively. TUNEL-positive neurons were counted for each condition.

## TDP43-BiFC measurements

HEK293 TDP43-BiFC cells were cultured in the same HEK293T cell media supplemented with 100 μg/mL Geneticin (G418). All cells were maintained in a humidified chamber with 5% $CO_2$ at 37 °C. HEK293 TDP43-BiFC cells were plated on a 96-well plate with an Opti-MEM medium (Gibco, #31985070). After 12 h, the cells were transfected with 0.1 μg of hsPEX19-WT or hsPEX19-FV plasmid using Lipofectamine®2000 reagent (Invitrogen, #11668027). At 13 h post-transfection, TDP43-BiFC cells were treated with Forskolin (30 μM) or Apicidin (1 μM). After 36 h, nuclei were counterstained with Hoechst 33342 (Invitrogen, #H1399). Fluorescence images were automatically acquired using Operetta CLS (PerkinElmer) with a 20× water immersion objective (TDP43-BiFC; $\lambda_{ex}$ = 460–490 nm and $\lambda_{em}$ = 500–550 nm, Hoechst; $\lambda_{ex}$ = 355–385 nm and $\lambda_{em}$ = 430–500 nm). The fluorescence intensities of TDP43-BiFC were quantified using Harmony v4.9 software (PerkinElmer). Data for each replicate were collected from 20 fields of view per well in the 96-well plate.

## Drosophila melanogaster stocks

The fly lines $W^{1118}$ (stock #5905), UAS-Httex1-20Q (stock #68412), UAS-Httex1-93Q (stock #68418), UAS-hsDNAJB1 (stock #82244), UAS-hsHSPA1A (stock #97467), Elav-Gal4 (stock #8765), and D42-Gal4 (stock #8816) were obtained from the Bloomington Drosophila Stock Center (USA). All flies were maintained at 27 °C.

To generate transgenic fly lines, N-terminally HA-tagged hsPEX19-WT and hsPEX19-FV genes were subcloned into the pACU2 vector (gift from Chun Han, Cornell University). The pACU2 vector lacking the HA-hsPEX19 gene was used as a negative control. The UAS-pACU2 vector, UAS-HA-hsPEX19-WT, and UAS-HA-hsPEX19-FV transgenic fly lines were

generated by BestGene, Inc. These *hs*PEX19 transgenic fly lines were crossed with Httex1 transgenic fly lines, and the genotypes of generated transgenic fly lines used in this study are listed in Supplementary Table 1.

## Climbing assay

Ten to fifteen male flies were collected and transferred into an acrylic cylinder (3 cm diameter, 18 cm height) without the use of any $CO_2$ anesthesia but with cotton-sealed. Prior to the climbing assay, the collected flies were transferred into a new food vial within 24 h. The flies were acclimatized for 20 min in the cylinder. The climbing ability was measured by tapping the cylinder against a table, which was recorded for 1 minute. A climbing index is the proportion of flies climbing > 5 cm from the bottom of the cylinder within 5 or 6 s. Seven technical trials were conducted for each individual experiment, and the average of these trials was considered as one biological replicate.

## Lifespan assay

A maximum of 15 male flies were collected within 24 h after pupal eclosion (APE) in a food vial and transferred to a new vial every two days. The number of dead flies was counted daily until all flies died. Lifespan data were plotted as Kaplan-Meier survival curves and statistical analyses were performed using both the log-rank (Mantel-Cox) test and the Gehan-Breslow-Wilcoxon test. The log-rank and Gehan-Breslow-Wilcoxon are more sensitive to distributional differences in the survival curves at a later time and an early time[99], respectively.

## Immunohistochemistry

Male fly heads were dissected to extract brains in Schneider's insect medium (Sigma, #S0146). The brains were fixed with 3.7% formaldehyde for 20 min at room temperature. After washing with the washing buffer (0.3% Triton X-100 in PBS) for 10 min (a total of six washes), the brain samples were incubated in the blocking buffer (5% Normal Donkey Serum in PBS containing 0.3% Triton X-100) for 1 h at room temperature with gentle shaking. To stain Httex1-93Q and HA-*hs*PEX19 proteins, the samples were incubated with Huntingtin (mEM48) mouse (1:200 dilution, Sigma-Aldrich, #MAB5374) and HA rabbit (1:200 dilution, Cell Signaling, # 3724S) primary antibodies at 4 °C overnight, respectively. After washing with the washing buffer, the samples were further incubated with rabbit Alexa Fluor 555 (1:200 dilution, Thermo Scientific, #A-21428) and mouse Alexa Fluor 488 (1: 200 dilution, Invitrogen, #A-11001) secondary antibodies for 2 h at room temperature. The stained samples were mounted onto slides using Antifade Mounting Medium with DAPI (VectorLabs, #H-1200-10). All stained brain samples were taken at 400× magnification using a 40× water immersion objective, acquired by Zeiss LSM700 confocal microscopy. Confocal microscopy images were set to a threshold to eliminate non-puncta fluorescence signals using Zeiss ZEN software. The same threshold settings were applied to all the images of the experiment. The total number of puncta in the region of interest was calculated using Fiji software.

## Preparation of Drosophila head lysates

*Drosophila* head lysates were prepared for the filter trap assay as described in the previous study with minor modifications[100]. 30 male fly heads were ground for 1 min in homogenization buffer (2% SDS, 1× Protease Inhibitor Cocktail, 1 μg/mL DNase). Lysates were centrifuged at $6900 \times g$ for 10 min at 4 °C. 75 μg of the clarified lysate was used for the filter trap assay and the Httex1-93Q aggregates were probed using MW8 antibody (1:1000 dilution, DSHB, #concentrate 0.1 mL).

## Statistics & reproducibility

Statistical analysis was conducted using GraphPad Prism Software. As indicated in Figure legends, we used the Log-rank (Mantel-Cox) test, Gehan-Breslow-Wilcoxon test, one-way ANOVA with Tukey post-hoc test, and two-way ANOVA with Tukey post-hoc test. All data were shown as mean ± SD or SEM and the exact numbers of samples were shown in the figure legends. *p*-values were represented with asterisks: $*p < 0.05$; $**p < 0.01$; $***p < 0.001$; and $****p < 0.0001$. No statistical method was used to predetermine the sample size, and no data were excluded from the analyses. For *Drosophila* experiments, flies of the same genotype were randomly selected for each experiment.

## Reporting summary

Further information on research design is available in the Nature Portfolio Reporting Summary linked to this article.

## Data availability

All data generated in this study are provided in the manuscript's main text (Figs. 1–5), Supplementary information (Supplementary Figs. 1–15), and Source Data file. A previously published structure of PEX19 used to show both mutation sites and Bpa incorporation positions is available under PDB code 5LNF. Source data are provided in this paper.

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

## Acknowledgements

We thank S.-h. Park, K. Shen, Y. Cho, and the members of Shan lab for valuable discussions on experimental methods. We thank the Research Solution Center and Korea Virus Research Institute at the Institute for Basic Science for the confocal microscopy facility and fluorescence plate reader, respectively. We thank W.-k. Huh, F.U. Hartl, J. Frydman, R. Shalgi, S.E. Radford, and S.-o. Shan for sharing the *W303a* yeast strain or plasmids. This work was supported by grants from the Institute for Basic Science (IBS-R030-Y1 to H.C. and IBS-R030-C1 to H.M.K.) and the National Research Foundation (NRF) grant funded by the Korea government (MSIT) (RS-2023-00261784 to Y.K.K. and 2022R1C1C100714612 to S.L.).

## Author contributions

H.C. conceived the study. H.C., S.B.L., and H.M.K. designed the study. J.O., C.C., E.S.K., K.W.M., H.C.J., H.K., M.K., S.H.A., N.L., M.G.J., H.S.B, and H.C. carried out all experiments. J.O., C.C., E.S.K., K.W.M., H.C.J., H.K., M.K., N.L., M.G.J., S.L., and H.C. performed data analysis. H.C., S.B.L., H.M.K., S.L., and Y.K.K. supervised the study. H.C., J.O., E.S.K., K.W.M., H.K., M.K., and S.L. drafted the manuscript. H.C. edited the manuscript with input from J.O., E.S.K., S.B.L., and H.M.K. All authors read and approved the final version of the manuscript.

## Competing interests

The authors declare no competing interests.
