## [Transparent Peer Review file · Nature Communications]

Engineering a membrane protein chaperone to ameliorate the proteotoxicity of mutant huntingtin

Corresponding Author: Dr Hyunju Cho

Version 0:

Reviewer comments:

Reviewer #1

(Remarks to the Author)

In this manuscript, the authors engineer the ATP-independent chaperone PEX19, which ordinarily targets peroxisomal membrane proteins to peroxisomes, to clear mutant huntingtin aggregates. The authors created a large library of random mutants that they screen in a yeast system to identify two variants. They go on to show that the variants prevent aggregation in vitro and in cellular models. They then probe the mechanism and find that a specific residue in the helix of the mutant binds to the N17 domain of huntingtin. This variant also rescues phenotypes in primary neurons and flies. Overall, this is a very interesting study that ranges from initial library design to protective phenotypes in relevant higher model systems and will be an impactful addition to the literature. However, in a number of areas the analysis appears to be too preliminary. In several instances key controls are missing, several key results show only representative data but should be quantified, and some additional follow up assays should be added to support the authors' claims. This is particularly important for the experiments where they conclude that the F255 residue specifically binds to the N17 domain of Httex. While their evidence supports the idea that F255 is involved, the findings are overstated. Elucidating the mechanism of these new variants is important for making this paper impactful, so additional experiments should be included to support this mechanism. Specific comments are described in more detail below. I think once addressed, this could be a very nice paper of significance to the field.

- The authors report that there were 90,000 transformants in the library, but it seems that this reflects transformation efficiency rather than number of unique variants in the library. This raises the question- are PEX19 variants that rescue rare or common? The authors should determine the approximate number of distinct variants that were screened.
- My understanding is that mutant m1 contained 8 mutations and m2 contained 7 mutations, and then the authors selected the two overlapping mutations to move forward with. However the writing of this section of the results is confusing. Also because yeast can take up multiple plasmids, it is possible that not all 7-8 mutations are in a single PEX19 gene. Could the authors explicitly state that these colonies had 7-8 mutations, and also state the uncertainty that these may or may not all be in a single gene?
- Engineered chaperones are often toxic in a cellular system, which can limit therapeutic use. Could the authors test if these mutants confer a temperature sensitive phenotype in yeast?
- The data in Figure 1d is convincing that the two mutants decrease aggregation. However, this microscopy data should be quantified, particularly because it appears that the F1 mutant has weaker activity here, but similar activity in the SDS-solubility assay (1E).
- In Figure 1F it seems that no statistical test was performed, though the authors describe the results as significant.
- The results in Figure 2C look nice, however they (and all of the filter trap assays throughout the paper) are missing a loading control. The samples should be applied to a nitrocellulose membrane in parallel to cellulose acetate to assay total protein. This is particularly important because some of the spots appear absent, and there is no way to know if there was soluble Httex in these samples, or just no Httex is present/applied to the membrane. Also the assays should be repeated in triplicate and the results quantified. This should also be done for the other filter trap assays throughout the paper. Quantitation is also important because the authors argue that the hsPEX19 is more effective than scPEX19, a comparison that cannot be made without quantifying the results.
- For the TEM in panel 2E it would be useful to either quantify the results or show multiple fields of view in the supplement to show that these fields are representative. Also it would be useful to add additional timepoints if possible. The F1 variant shows a mixed population which may be due to inhibition being incomplete or inhibition being insufficient over time. If a 24h timepoint is included would F1 still inhibit, and would the FV variant still fully inhibit?

- A general issue is that the in vitro inhibition assays are only assessed by filter trap and EM and both are performed qualitatively. It would be useful to include ThioflavinT assays to monitor aggregation over time. The authors frequently refer to the inhibition occurring at “early stage” but without being able to visualize the lag, nucleation, and rapid growth stages these statements are overreaching. This is also important because in panel 1A WT PEX19 does not rescue but in Fig2C-D they see complete inhibition to 3h. So there is some activity and perhaps a new species is generated at ~3h.
- As described below for the neuronal assays, the data in 2F should be quantified. It would also be useful to perform toxicity assays to determine if suppression of aggregation is correlated with toxicity.
- The data in Figure 3B are nice, showing that ΔN aggregates cannot be countered by PEX19. However, as shown, this is negative data. To make a convincing argument it is important that the positive control with full length Htt is performed alongside these assays and shown on the same blot/figure panel.
- The data showing that hsPEX19 does not suppress ataxin3 aggregation is nice evidence suggesting that the N17 domain is the main recognition site. I would suggest moving this data from the supplement to the main text. This also gets at mechanism, which is otherwise somewhat limited. Could the authors add a control where the N17 domain is fused to ataxin3 to determine if PEX19 now inhibits ataxin3 aggregation?
- It is interesting that the authors are able to pinpoint the specific region of PEX that interacts with N17 in Figure 3. However, all of this data consists of pull down assays and this interaction may be transient. The authors should measure a dissociation constant, K_d , for this interaction. Corroborating this direct interaction is important, as it is a key finding in the paper.
- In Figure 4E PEX26 pulls down Httex, but in panel 4G there is no cross linking, which is surprising. The authors explain this by saying that the particular residue does not interact, but if a pull down occurs, this should be sufficient to promote cross linking. To make this argument the authors should corroborate these results with a complementary technique. Alternatively perhaps by placing the cross linker in a different position they might see cross linking.
- Also along these lines, a major focus of the paper is the conclusion that the Phe in the $\alpha 4$ helix of hsPEX19 directly interacts with the N17 domain. All evidence for this comes from XL-MS. The authors should use a complementary approach to validate this interaction. As shown, this is inconclusive on a direct interaction with this specific residue, it just suggests the region plays an important role.
- In Figure 5A, for the PEX19-WT panel, while the top right corner appears to show fragmentation, the region outside of the inset appears quite similar to that of the PEX19-FV sample. Therefore this data should be quantified. It would also be useful to perform a toxicity assay, either here or with the HEK cell assays.
- In the neuronal and fly assays just the FV mutant is tested. Presumably testing both the FV and FI mutant would be too difficult, which is understandable. However, the authors should explain why they chose FV rather than FI.
- In the introduction the engineering of the AAA+ disaggregase Hsp104 is highlighted. In addition, other recently discovered disaggregases (including VCP, DAXX, KaryopherinB2, and HTRA1) should be included, particularly given that several of these are also ATP-independent and do not require co-factors.
- Also can the authors comment more in the introduction about how they chose PEX19 in the first place? There are many similar genes they could have selected.

Reviewer #2

(Remarks to the Author)

Engineering a membrane protein chaperone to ameliorate the proteotoxicity of mutant huntingtin
Oh et al.

In this study, protein variants of the cytosolic chaperone PEX19 (e.g., hsPEX19-FV) were engineered that can slow down mutant HTT exon-1 (mHTT_{ex1}) aggregation in cell-free and cell-based assays as well as in HD transgenic flies. The authors show that hsPEX19 variants directly associate with the N17 domain of mHTT_{ex1}, suggesting that they bind to soluble mHTT_{ex1} and slow down spontaneous aggregation because they interfere with an early event in the polymerization process. Importantly, they show that hsPEX19-FV-mediated effects on mHTT_{ex1} aggregation in HD models are associated with reduced toxicity, supporting the hypothesis that mHTT_{ex1} aggregation and proteotoxicity are linked. This is an interesting mechanistic study with high relevance for the protein misfolding/aggregation field. It should be published in Nature Communications, when a few key points have been specifically addressed by the authors.

The authors state in their abstract and also in other parts of this manuscript that PEX19 protein variants can prevent mHTT_{ex1} aggregation in HD models. This in my view is an overstatement and should be changed. The data indicate that e.g., hsPEX19-FV can slow down spontaneous mHTT_{ex1} aggregation in cells but does not prevent the formation of SDS-stable aggregates (see e.g., Fig. 2 g,h and extended data Fig. 7c, f). In vitro assays (see data Fig. 2c and d): It is not sufficient to show only data points for 0, 3 and 6 h. Fibrillar SDS-stable mHTT_{ex1} aggregates might form after 12, 24 or 48h in cell-free assays.

The authors present evidence that PEX19 protein variants interact with the N17 domain and they use cross-linking to show this association (Fig. 4). It would be important to know whether the interaction between PEX19-FV and mHTT_{ex1} is strong or weak. The authors should provide a dissociation constant (K_d) for this interaction to substantiate their PPI results.

The data generated with the HD fly strains are not fully convincing. The authors e.g., state that hsPEX-FV co-expression does not change the survival of HTT_{ex1}-20Q flies. This, however, is not shown in Fig.5E. Complete survival curves need to be presented for all the control strains (Fig. 5D and E). Also, complete time-courses need to be presented for climbing assays (control flies and HD transgenic flies). The data presented for 12-day-old flies are interesting (Fig. 5B). However, it would be important to know whether at later time points (e.g., 20 or 30 days) climbing is also improved by the PEX19-FV protein variant. Finally, filter trap assays (similar to Fig.2c) should be performed with fly head extracts to show the impact of PEX19-

FV on mHTTex1 aggregation in vivo. The results presented in Extended Data Fig. 7 are not fully convincing. The detected puncta may be vesicular structures rather than mHTTex1 aggregates. Additional data are needed to substantiate the relationship between mHTTex1 aggregation and survival of HD transgenic flies.

Minor points

Fig. 2 (e): Data for FI also fits the results of filter trap assay in Extended Data Fig. 3 (c). Mutation FV is more effective in inhibiting the aggregation than mutation FI.

Fig. 3 (d,e): Check legend > hsPE+X19-FVBpa? For (d), what is presented by the band below the complex PEX19Bpax51Q? Truncated PEX19? Pattern matches the pattern of PEX19 only. How is PEX19 getting truncated?

Supplementary Table 1: Not provided.

Extended Data Fig. 1 (c,d,e): What is the unit? Bars do not match blots. Blot shows the opposite result. Were 25Q and 97Q analyzed separately? For (e), which band was analyzed (upper or lower)?

Extended Data Fig. 4 (e): The GFP channel for hsPEX19-WT looks similar to the vector control. Is that anticipated?

Extended Data Fig. 6 (d,e,f,g): EV is the abbreviation for what?

Reviewer #3

(Remarks to the Author)

The manuscript by Oh et al seeks to engineer PEX19, involved in the targeting of peroxisomal proteins, towards the polyQ expansion-containing mHttex1 proteins involved in Huntington's disease. The authors identified a double mutant of Pex19, using a yeast model of mHttex1 toxicity, which shows significantly improved anti-aggregation activity towards mHttex1 compared to wildtype Pex19. The alteration in activity is transferrable to human PEX19, and mechanistic studies support a new interaction between the engineered PEX19 and the N17 motif in mHttex1 that is responsible for improved activity. The efficacy of the engineered PEX19 was tested in a Drosophila model, which showed modest rescues of the toxicity of Httex1-93Q overexpression. The study represents a successful first step in an interesting engineering effort to generate molecular chaperones targeting disease-linked protein aggregation problems. As a rather new approach, it presents some difficulty in evaluation. In particular, I struggled on the bigger context and impact of this study, as it is neither a tour de force engineering effort nor a fundamental study. In the end, the most significant lessons for me are the following: the study demonstrates that it is fairly straight-forward to significantly improve/generate a new chaperone function with only a few mutations, and that by targeting unique motifs in a target protein (such as N17 in mHttex1), it is possible to generate client specificity in the chaperone. I do have a number of reservations on how strong the manuscript demonstrated these points and how effective the chosen approach is, which are detailed below.

1. What is the rationale for choosing Pex19, which has limited interaction with mHtt, as the starting point for engineering, rather than improving on chaperones that naturally have strong activity towards Htt? I think that there is a case against using Hsp104, which is a large yeast protein with limited client specificity and immunogenic potential. However, other mammalian chaperones such as NAC, Hsp70 and J proteins can be candidates. What is the advantage of Pex19 over these alternative options?

2. Did the engineered mutations interfere with peroxisome targeting? Does overexpression of the mutant Pex19 interfere with wildtype Pex19 function? These are confusing points throughout the manuscript. Farnesylation is important for Pex19 function, and overexpression of the engineered Pex19 drastically reduced Pex19 farnesylation levels. One can infer from this that either the mutants are not farnesylated (and therefore not active in PMP targeting), or that their expression inhibits Pex19 farnesylation. However, the authors assert that the mutants is unaltered in mediate PMP targeting, but the only evidence is that unfarnesylated mutant Pex19 co-purifies with the substrate Pex26. Clarifying this point is important in assessing the approach and the activity/specificity of engineered chaperone mutants.

3. Most amyloid diseases are strongly seed-catalyzed, such as amyloid beta in Alzheimer's and alpha-synuclein in Parkinson's. Is the aggregation of mHtt also seed-catalyzed? If so, does the engineered Pex19 protect against seeded aggregation of mHtt, which would be closer to conditions in a diseased patient?

4. How does the efficacy of the engineered Pex19 compare with native mammalian chaperones that have been reported to antagonize mHtt aggregation, such as NAC and DNAJB6, in vitro and in the Drosophila model? In particular, how does the effect of overexpressing these chaperones compare with expressing mutant Pex19? A higher activity of the mutant Pex19 will strongly support the engineering approach.

5. For the primary neuron studies (Fig. 5A):

- At what level is PEX19 overexpressed compared to native levels?

- The neurons look abnormal when PEX19 is overexpressed. The nuclear area appears much smaller (was the map2 channel saturated?). The images appear to be an overcrowded area with too many overlapping neurons. The authors should provide clearer images, preferably with quantification of the axon morphology.

6. For the Drosophila studies:

- The aggregation of the Httex1 proteins from these neurons should also be analyzed.

- It appears that overexpression of PEX19 (WT and mutant) decreases the lifespan of otherwise healthy flies, but this point was skipped over.

- I don't understand the statistics: in 5E, the difference between flies with and without PEX19 overexpression is pretty large (a horizontal difference of ~ 8 days), but was labeled * or ns. In 5F, the difference is at most two days, but was given ****.

7. An important point of this study is that the mutant Pex19 is specific for mHttex1 Huntington, based on comparison with TDP43 and another polyQ protein. The specificity will be more convincingly demonstrated from an IP-MS approach to test if

the engineered and overexpressed PEX19 show enhanced interaction with other cellular proteins besides mHttEx1.

Version 1:

Reviewer comments:

Reviewer #1

(Remarks to the Author)

I think the authors did a very nice job of addressing my concerns. With the newly added data and analyses, I think the manuscript is improved and appropriate for publication in Nature Communications.

Reviewer #2

(Remarks to the Author)

Oh et al/NCOMMS-24-26029A

Engineering a membrane protein chaperone to ameliorate the proteotoxicity of mutant huntingtin

The authors have now comprehensively addressed my concerns. The paper is now suitable for publication in Nature Comm. I have no further comments. This is a well written paper that is of high importance for the scientific community working on HD.

Reviewer #3

(Remarks to the Author)

The authors have addressed most of my questions on the previous version of the manuscript. Before acceptance, though, the authors need to address a few minor issues:

1. Supplementary figure 6: The effect of seed in accelerating HttQ aggregation is very modest, and the aggregation observed in the presence of 100 nM seed is likely / probably largely driven by unseeded reaction. The author should tone down the conclusion that the chaperone prevents seed-catalyzed Htt aggregation.

2. Supplementary figure 8: In the SPR data, neither the association nor dissociation time traces have reached completion to accurately determine the rate constants. The dissociation trace is particularly problematic (k_{off} is 10^{-4} s^{-1} , or $\tau = 10000 \text{ s}$, but the trace only lasted 600 s). In fact, if one uses the magnitude of binding at each concentration as a proxy of equilibrium, the binding has not fully saturated at 100-250 nM. I think the SPR data provide evidence for high nanomolar to sub-micromolar affinity binding to HttQ51 that is specific for the N17 motif, but the K_d of 2.5 nM is likely incorrect. The authors should rephrase the conclusion to reflect the uncertainty.

3. Line 318: reference to Fig. 4 i-k for PEX19 xlink to PEX26. Figures 4j and 4k are with Htt51Q, not PEX26.

We greatly appreciate the reviewers' comments and suggestions, which have clarified and significantly improved the manuscript. We have now addressed the reviewers' concerns through the addition of significant new data alongside textual revisions. Our point-by-point responses (**blue**) to reviewers' comments (**black**) are shown below.

Reviewer #1 (Remarks to the Author):

In this manuscript, the authors engineer the ATP-independent chaperone PEX19, which ordinarily targets peroxisomal membrane proteins to peroxisomes, to clear mutant huntingtin aggregates. The authors created a large library of random mutants that they screen in a yeast system to identify two variants. They go on to show that the variants prevent aggregation in vitro and in cellular models. They then probe the mechanism and find that a specific residue in the helix of the mutant binds to the N17 domain of huntingtin. This variant also rescues phenotypes in primary neurons and flies. Overall, this is a very interesting study that ranges from initial library design to protective phenotypes in relevant higher model systems and will be an impactful addition to the literature. However, in a number of areas the analysis appears to be too preliminary. In several instances key controls are missing, several key results show only representative data but should be quantified, and some additional follow up assays should be added to support the authors' claims. This is particularly important for the experiments where they conclude that the F255 residue specifically binds to the N17 domain of Httex. While their evidence supports the idea that F255 is involved, the findings are overstated. Elucidating the mechanism of these new variants is important for making this paper impactful, so additional experiments should be included to support this mechanism. Specific comments are described in more detail below. I think once addressed, this could be a very nice paper of significance to the field.

1. The authors report that there were 90,000 transformants in the library, but it seems that this reflects transformation efficiency rather than number of unique variants in the library. This raises the question- are PEX19 variants that rescue rare or common? The authors should determine the approximate number of distinct variants that were screened.

When we generated the plasmid library using Gibson assembly, we transformed the Gibson assembly reaction and then counted the approximate number of colonies ($\sim 2.5 \times 10^5$ colonies). We randomly took 80 colonies from the plates and performed colony PCR to identify mutation sites. As shown in the table below, each colony has a different mutation site. We pooled out all $\sim 2.5 \times 10^5$ colonies and purified plasmid DNA. Assuming the colonies have distinct variants, we estimated that the plasmid library size is approximately 2.5×10^5 . As shown in **Table-response 1**, error-prone PCR often introduced either a premature stop codon (denoted as “*”) or a frameshift in the yeast Pex19 gene. Since frameshifts often generate a “null mutation” that does not have any stop codon, we thought determining the actual PEX19 library size could be complicated. Nevertheless, the library size is ~ 2.7 -fold larger than the number of yeast transformants, and thus, the identified yeast colonies are likely to have different PEX19 variants.

Table-response 1. The list of yeast PEX19 mutations in each colony.

The DNA sequences of a total of 80 colonies were analyzed, and each mutation site was separated by “/” in the table.

Number of mutated residues	
1	Y6N/ Q114*/ N85S/ K151N/ S121-Frameshift/ D282G/ P298A/ H308L
2	N2I, T21-Frameshift/ Y6I, K44*/ K22T, M232T/ A26T, Q57*/ D29E, K34*/D30E, Q32*/ V31E, I294P/E45V, K69N/ E49D, K281*/ P66L, E152-Frameshift/ P66S, Q259*/ E78V, T285I/ N141I, K151*/ E227G, I266T/ I262T, E265V
3	L16F, D17R, E18*/K22I, S41N, E67*/E25V, N43S, L185I/ D30N, K46N, Q76*/ G35S, I212L, H249R/ S41C, K44I, D54-I321-del/ E42G, K123R, S171I/ N47K, K166M, H308L/ A59T, S234R, P298T/ A59P, N81S, E320V/ M71I, A107T, L287V/ F79I, S160T, A197V/ A80R, D210Y, M219V/ N81I, K257T, L291V/ K93M, A108T, L291-Frameshift/ P137R, F240L, T285I/ S168P, D209E, D210V
4	E3G, I180K, G244C, E245V/ D7G, N91Y, A190S, E227*/ N8D, N91I, I177T, G192-Frameshift/ D17E, D40V, K214I, K230*/ D19E, N145I, K156E, K204I/ P20L, K69R, E105D, G138*/ T21S, S50G, I142V, Q259R/ P28S, V73I, Q187L, S191-Frameshift/ N39K, N60D, N77D, L147Q/ S53I, M72V, N246D, E320*/ E67D, S120C, N125D, N155S/ L82M, S121T, K123E, L150-Frameshift/ N85I, N174del, V238I, E311V/ A106T, K123E, K204T, M219K/ S102G, A108T, I266N, G293*/ N141H, L185S, N243Y, L288F/ L184F, M188K, T220M, V284I
5	N2Y, D54V, A108D, A162T, T164S/ A48T, S121N, S196N, L203R, E207K/ K51N, S121N, N125D, D264E, E311V/ V73L, N92D, Q187L, A211T, E265K/ E78K, K166E, G192R, E235G, Q293*/ S160F, Q200R, I262V, L294V, G295D
6	N4T, L15F, A80I, T94I, E207K, Y270*/ D11V, T159I, R233C, D296G, S301C, E311G/ S36F, K99T, M219I, K257E, M316I, I321N/V37L, E89D, E106V, K129N, I253K, L319S/ N88Y, E235K, F236Y, G237D, H308L, L319M
7	E5K, D17G, K69E, L161P, K194R, T255S, D322V/F9Y, D40E, N77S, A103V, P137A, L291I, D322E/ K22E, T127A, S135I, S182P, D186E, D264Y, K307E/ E89V, A103D, K123R, S126C, T127A, N155D, G193*/ E61G, D74G, K84I, N145Y, R149M, Y275C, E314D/ Q76L, A107V, N125Y, S126N, I177T, L181S, D289G
8	N4I, K69N, M71V, N81D, N153Y, T165A, N243S, P305S
9	A48T, E63G, E70G, D186V, K222E, E247G, K257T, V263L, Y275*/ K51N, V56I, K99R, V110A, T146R, M188I, A218L, I262F, S301R
10	L15I, N60I, L68S, M72L, C119Y, N133K, R149A, L161I, N199D, K279R
11	L15P, S41T, A48D, E49G, I101T, G154D, N155Y, R170L, E235D, E271V, S301C
12	N4S, K46M, Q57R, A107V, T126N, T131A, N145K, V157A, E198V, K214I, P228R, R133C, H249D, E251-Frameshift

2. My understanding is that mutant m1 contained 8 mutations and m2 contained 7 mutations, and then the authors selected the two overlapping mutations to move forward with. However the writing of this section of the results is confusing. Also because yeast can take up multiple plasmids, it is possible that not all 7-8 mutations are in a single PEX19 gene. Could the authors explicitly state that these colonies had 7-8 mutations, and also state the uncertainty that these may or may not all be in a single gene?

Since yeast often takes up multiple plasmids, we carefully assessed the sequencing results of colony PCR during the yeast screening experiments. If yeast cells harbor two plasmids, the chromatogram should show a mixture of two nucleotide peaks (PMID: 25407485). As shown in **Supplementary Fig. 2a, b**, the chromatograms have single peaks throughout the whole PEX19 sequence, including the specific mutation sites (highlighted in the dotted boxes in **Supplementary Fig. 2a, b**). Thus, we conclude that the selected two colonies highly likely contain single plasmids with 7 or 8 mutations. We have added this point to the Results section (page 5).

3. Engineered chaperones are often toxic in a cellular system, which can limit

therapeutic use. Could the authors test if these mutants confer a temperature sensitive phenotype in yeast?

We thank the reviewer for this suggestion. In contrast to the identified Hsp104-A503V variant (PMID: 37625404 and PMID: 24439375), scPEX19 variants did not exhibit a temperature-sensitive phenotype in yeast, indicating that those scPEX19 variants are not toxic at either 30°C or 37°C (see **Supplementary Fig. 1c**).

4. The data in Figure 1d is convincing that the two mutants decrease aggregation. However, this microscopy data should be quantified, particularly because it appears that the FI mutant has weaker activity here, but similar activity in the SDS-solubility assay (1E).

Since Western blot analysis uses whole yeast cell lysates, we initially thought that this analysis represents the total aggregate status of 97QΔP in the yeast cells. Thus, we quantitatively assessed the SDS-insoluble 97QΔP fractions using Western blot and qualitatively assessed microscopy data. These approaches showed that both scPEX19-FV and scPEX19-FI reduced the cellular aggregation of 97QΔP in yeast. As suggested, we further quantified the microscopy data (new **Fig. 1e**). Since the statistical test showed that there is no significant difference between scPEX19-FV and scPEX19-FI, we have replaced the original with better representative images for these data in **Fig. 1d**.

5. In Figure 1F it seems that no statistical test was performed, though the authors describe the results as significant.

We reported the results of the statistical analysis in the figure (now in **Fig. 1g**). As shown in **Fig. 1g**, both scPEX19-FV and scPEX19-FI significantly reduced the aggregation of Httex1-97QΔP proteins compared to scPEX19-WT.

6. The results in Figure 2C look nice, however they (and all of the filter trap assays throughout the paper) are missing a loading control. The samples should be applied to a nitrocellulose membrane in parallel to cellulose acetate to assay total protein. This is particularly important because some of the spots appear absent, and there is no way to know if there was soluble Httex in these samples, or just no Httex is present/applied to the membrane. Also the assays should be repeated in triplicate and the results quantified. This should also be done for the other filter trap assays throughout the paper. Quantitation is also important because the authors argue that the hsPEX19 is more effective than scPEX19, a comparison that cannot be made without quantifying the results.

We added the loading controls for all filter trap assays and now show quantified results in triplicate in the main and supplementary figures.

We would like to clarify one thing here. We never argued that the hsPEX19 variants are more effective than scPEX19 variants in the manuscript. As *Reviewer 3* also stated, due to the immunogenic potential of yeast proteins, we initially aimed to convert the yeast

scPEX19 variants to *hsPEX19* variants but, in the manuscript, only emphasized that the two conserved residues in the $\alpha 4$ helix of PEX19 are important for substrate recognition. We have now added this point in the revised manuscript (page 6).

7. For the TEM in panel 2E it would be useful to either quantify the results or show multiple fields of view in the supplement to show that these fields are representative. Also it would be useful to add additional timepoints if possible. The FI variant shows a mixed population which may be due to inhibition being incomplete or inhibition being insufficient over time. If a 24h timepoint is included would FI still inhibit, and would the FV variant still fully inhibit?

As the reviewer suggested, we have added multiple TEM images (8 images for 24 h sample and 6 images for 15 h sample) to **Fig. 2f** and **Supplementary Fig. 5a, b**. The TEM images show that *hsPEX19-FV* fully inhibited Httex1-51Q fibril formation at 24 h. In addition, the results of the Thioflavin T assay showed that *hsPEX19-FV* completely suppressed fibril formation of Httex1-51Q up to 47 h (**Fig. 2e**). In contrast, *hsPEX19-FI* delayed fibril formation by Httex1-51Q but was insufficient for complete inhibition of Httex1-51Q fibril formation at 47 h (**Fig. 2e-f**).

8. A general issue is that the in vitro inhibition assays are only assessed by filter trap and EM and both are performed qualitatively. It would be useful to include ThioflavinT assays to monitor aggregation over time. The authors frequently refer to the inhibition occurring at “early stage” but without being able to visualize the lag, nucleation, and rapid growth stages these statements are overreaching. This is also important because in panel 1A WT PEX19 does not rescue but in Fig2C-D they see complete inhibition to 3h. So there is some activity and perhaps a new species is generated at ~3h.

We thank the reviewer for this suggestion. We have now additionally performed Thioflavin T assays to monitor Httex1 fibril formation kinetics (up to 47h). As shown in **Fig. 2e**, *hsPEX19-FV* inhibits the early aggregation of Httex1-51Q.

9. As described below for the neuronal assays, the data in 2F should be quantified. It would also be useful to perform toxicity assays to determine if suppression of aggregation is correlated with toxicity.

We have now added quantification of % cells containing 134Q aggregates to the confocal microscopy images (**Fig. 2g, h**). The data from the confocal microscopy analyses are highly correlated with the data from the filter trap assay (**Fig. 2i**).

Since primary neurons are more pathologically relevant, we performed a toxicity assay using primary neurons instead of HEK293T cells (see **Supplementary Fig. 12a, b**).

10. The data in Figure 3B are nice, showing that ΔN aggregates cannot be countered by PEX19. However, as shown, this is negative data. To make a convincing argument it is

important that the positive control with full length Htt is performed alongside these assays and shown on the same blot/figure panel.

As the reviewer suggested, we have added data showing Httex1-51Q-WT and Httex1-51 detected on the same blots (**Fig. 3b, c**). Along with the data on Ataxin (**Fig. 3d-g**, also see below), these data suggest that the N17 domain is the primary recognition site of *hsPEX19* variants.

11. The data showing that *hsPEX19* does not suppress ataxin3 aggregation is nice evidence suggesting that the N17 domain is the main recognition site. I would suggest moving this data from the supplement to the main text. This also gets at mechanism, which is otherwise somewhat limited. Could the authors add a control where the N17 domain is fused to ataxin3 to determine if *PEX19* now inhibits ataxin3 aggregation?

We thank the reviewer for the suggestion. We tagged the N17 domain at the N-terminus of ATX3-78Q (**Fig. 3d**). In contrast to ATX3-78Q-WT, all *PEX19* proteins reduced aggregation of N17-ATX3-78Q to a similar extent (**Fig. 3e, g**). As discussed in the main text (page 9), a recent study suggested that structural coupling between the N17 and polyQ repeat domains stabilizes the helical content of Httex1 and accelerates aggregation (PMID: 36864173). Thus, we speculate that *hsPEX19* variants suppress the aggregation of polyQ proteins more effectively when N17 is coupled to the polyQ-repeat domain.

12. It is interesting that the authors are able to pinpoint the specific region of *PEX* that interacts with N17 in Figure 3. However, all of this data consists of pull down assays and this interaction may be transient. The authors should measure a dissociation constant, K_d , for this interaction. Corroborating this direct interaction is important, as it is a key finding in the paper.

Since the Httex1-51Q protein itself is highly aggregation-prone, we used GST-Httex1-51Q for the binding affinity measurements. In our initial trials in the BLI (Biolayer interferometry) experiments, we found that immobilization of GST-Httex1-51Q in the sensor tip interfered with binding to *hsPEX19*-FV, potentially due to non-specific binding of GST-Httex1-51Q on the sensor surface. Thus, we immobilized *hsPEX19*-FV on the sensor and further analyzed the binding affinity of GST-Httex1-51Q-WT and GST-Httex1-51Q- Δ N to *hsPEX19*-FV (**Supplementary Fig. 8a, b**). GST-Httex1-51Q-WT binds to *hsPEX19*-FV with a high binding affinity ($K_D = 2.5 \pm 0.26$ nM), whereas no binding was observed with GST-Httex1-51Q- Δ N (**Supplementary Fig. 8a, b**). Consistent with data from both Filter trap assay and Thioflavin T assay, the dissociation rate of GST-Httex1-51Q-WT is slow enough ($k_{off} \sim 1 \times 10^{-4} \text{ s}^{-1}$) to act as a holdase for mHttex1

13. In Figure 4E *PEX26* pulls down Httex, but in panel 4G there is no cross linking, which is surprising. The authors explain this by saying that the particular residue does not interact, but if a pull down occurs, this should be sufficient to promote cross linking. To make this argument the authors should corroborate these results with a

complementary technique. Alternatively perhaps by placing the cross linker in a different position they might see cross linking.

In the His-PEX19 pulldown assay (now **Fig. 4g-h**), we showed that PEX19-WT and its variants interact with PEX26. We previously showed that the PEX19-M179^{Bpa} residue (located in the α 1 helix) crosslinked to PEX26 (PMID: 38585659), arguing that the α 1 helix is a primary binding site for PMPs (PMID: 28281558 & PMID: 25517356). As shown in **Supplementary Fig. 10b, c**, we further showed that both PEX19-M179^{Bpa}-FV and PEX19-M179^{Bpa}-FI crosslinked with PEX26. In addition, the zero-length crosslinker Bpa is known to crosslink with any residue within a radius of 3.1 Å centered on the ketone oxygen of Bpa (PMID: 8180191). This crosslinking length is much shorter than most chemical crosslinkers (i.e., BS₃ and DSS (spacer arm length 11.4 Å)). Combining all the data in **Fig. 4** and **Supplementary Fig. 10**, our results suggest that the α 4 helix of PEX19-FV serves as a specific binding site for the N17 domain of Httex1-51Q, whereas the α 1 helix of PEX19 serves as a primary binding site for PMPs.

14. Also along these lines, a major focus of the paper is the conclusion that the Phe in the α 4 helix of hsPEX19 directly interacts with the N17 domain. All evidence for this comes from XL-MS. The authors should use a complementary approach to validate this interaction. As shown, this is inconclusive on a direct interaction with this specific residue, it just suggests the region plays an important role.

Since there might be some confusion in the Bpa crosslinking assay, we added a description of the Bpa crosslinking assay in **Supplementary Fig. 9a**. Unlike other chemical crosslinkers (amine-reactive crosslinkers), the Bpa crosslinking assay is a site-directed crosslinking approach and as shown in **Supplementary Fig. 9a**, Bpa located at F255 covalently reacts with any residue within a radius of 3.1 Å centered on the ketone oxygen of Bpa (PMID: 8180191). Thus, we did not use the XL-MS approach to identify the crosslinking regions. Combining all data in **Fig. 3h, 3j, 4c, and 4d**, we concluded that the F255 residue in *hs*PEX19-FV directly interacts with the hydrophobic residues in the N17 domain of Httex1.

15. In Figure 5A, for the PEX19-WT panel, while the top right corner appears to show fragmentation, the region outside of the inset appears quite similar to that of the PEX19-FV sample. Therefore this data should be quantified. It would also be useful to perform a toxicity assay, either here or with the HEK cell assays.

As *Reviewer 3* also suggested, we quantified the degree of fragmentation (4 fragmentation scores) in one isolated neuron and summarized the results in **Fig. 5b**. While *hs*PEX19-WT showed various degrees of fragmentation, *hs*PEX19-FV effectively protected the neuronal fragmentation caused by the expression of Httex1-134Q (**Figs. 5a and b**).

In addition, we performed a TUNEL cell death assay and found that *hs*PEX19-FV reduced Httex1-134Q-associated toxicity in striatal neurons (**Supplementary Fig. 12a, b**).

16. In the neuronal and fly assays just the FV mutant is tested. Presumably testing both the FV and FI mutant would be too difficult, which is understandable. However, the authors should explain why they chose FV rather than FI.

As shown in **Figs. 2c-f**, our *in vitro* results suggest that *hsPEX19-FV* exhibits a higher chaperone activity for Httex1-51Q than *hsPEX19-FI*. Thus, we tested *hsPEX19-FV* in both primary neurons and drosophila models. We have added this point in the Results section (page 12).

17. In the introduction the engineering of the AAA+ disaggregase Hsp104 is highlighted. In addition, other recently discovered disaggregases (including VCP, DAXX, KaryopherinB2, and HTRA1) should be included, particularly given that several of these are also ATP-independent and do not require co-factors.

We thank the reviewer for the suggestion. We have added this literature information in the Introduction section (page 3-4).

18. Also can the authors comment more in the introduction about how they chose PEX19 in the first place? There are many similar genes they could have selected factors.

There is growing evidence that the dysfunction of peroxisomes observed in peroxisome biogenesis disorders is linked to neurodegenerative diseases. Furthermore, a previous study showed that *PEX19^{-/-}* adult flies exhibit a defect in climbing ability, suggesting the potential functional relevance of PEX19 to neurodegeneration (PMID: 29282281). Along with the fact that PEX19 does not require any cochaperone, cofactors, or complex assembly steps for its optimal chaperone activity, we hypothesized that PEX19 would be an ideal chaperone template to develop an engineered chaperone for mHttex1 aggregation (also refer to our response for the comment *Reviewer 3.1*). We have added this point to the Introduction section (page 4).

Reviewer #2 (Remarks to the Author):

Engineering a membrane protein chaperone to ameliorate the proteotoxicity of mutant huntingtin
Oh et al.

In this study, protein variants of the cytosolic chaperone PEX19 (e.g., *hsPEX19-FV*) were engineered that can slow down mutant HTT exon-1 (mHTT_{ex1}) aggregation in cell-free and cell-based assays as well as in HD transgenic flies. The authors show that *hsPEX19* variants directly associate with the N17 domain of mHTT_{ex1}, suggesting that they bind to soluble mHTT_{ex1} and slow down spontaneous aggregation because they

interfere with an early event in the polymerization process. Importantly, they show that hsPEX19-FV-mediated effects on mHTTex1 aggregation in HD models are associated with reduced toxicity, supporting the hypothesis that mHTTex1 aggregation and proteotoxicity are linked.

This is an interesting mechanistic study with high relevance for the protein misfolding/aggregation field. It should be published in Nature Communications, when a few key points have been specifically addressed by the authors.

1. The authors state in their abstract and also in other parts of this manuscript that PEX19 protein variants can prevent mHTTex1 aggregation in HD models. This in my view is an overstatement and should be changed. The data indicate that e.g., hsPEX19-FV can slow down spontaneous mHTTex1 aggregation in cells but does not prevent the formation of SDS-stable aggregates (see e.g., Fig. 2 g,h and extended data Fig. 7c, f). In vitro assays (see data Fig. 2c and d): It is not sufficient to show only data points for 0, 3 and 6 h. Fibrillar SDS-stable mHTTex1 aggregates might form after 12, 24 or 48h in cell-free assays.

Based on our new results in **Fig. 2e**, the equimolar concentration of *hsPEX19-FV* is sufficient to completely suppress large fibril formation by Httex1-51Q up to 47 h. Despite this high chaperone activity of *hsPEX19-FV*, assessing its expression levels relative to mHttex1 in cellular and *Drosophila* HD models would be difficult, thus making data interpretation complicated. Nevertheless, we agree with the reviewer that it might not be possible for *hsPEX19-FV* to completely prevent mHttex1 aggregation at later time points (>47 h) or in more complicated model systems. Thus, we have edited those sentences throughout the manuscript.

2. The authors present evidence that PEX19 protein variants interact with the N17 domain and they use cross-linking to show this association (Fig. 4). It would be important to know whether the interaction between PEX19-FV and mHTTex1 is strong or weak. The authors should provide a dissociation constant (K_d) for this interaction to substantiate their PPI results.

Please refer to our response above (*Reviewer 1.12*). As also shown in **Supplementary Fig. 8a**, GST-Httex1-51Q-WT binds to *hsPEX19-FV* with a high binding affinity ($K_D = 2.5 \pm 0.26$ nM).

3.1. The data generated with the HD fly strains are not fully convincing. The authors e.g., state that *hsPEX-FV* co-expression does not change the survival of HTTex1-20Q flies. This, however, is not shown in Fig.5E. Complete survival curves need to be presented for all the control strains (Fig. 5D and E).

We apologize for any confusion caused by statistical analysis with incomplete survival curves. As suggested, we have now included the complete survival curves for all control *Drosophila* lines (**Fig. 5e-f**) and conducted statistical analyses with two types of tests. The log-rank test is generally known to be more sensitive to distributional differences in the survival curves at a later time, whereas the Wilcoxon test tends to detect differences at an early time (METRON 68, 111–125 (2010). <https://doi.org/10.1007/BF03263529>).

As shown in **Fig. 5e-f**, we have reported statistical analysis results with these two tests, and we have revised the manuscript accordingly (page 13).

3.2. Also, complete time-courses need to be presented for climbing assays (control flies and HD transgenic flies). The data presented for 12-day-old flies are interesting (Fig. 5B). However, it would be important to know whether at later time points (e.g., 20 or 30 days) climbing is also improved by the PEX19-FV protein variant.

We appreciate the reviewer's comment regarding the climbing assay at later time points. Due to the marked increase in lethality observed in HD flies at later stages, we did not perform the climbing assay beyond 12 days. Our concern was that the increased number of sick survivors co-expressing *hsPEX19-FV* and *Httex1-93Q*, who would otherwise have succumbed to *Htt* toxicity, might introduce bias into the statistical analysis. Nevertheless, in light of the reviewer's suggestion, we conducted the climbing assay with 15-day-old flies (pan-neuronal expression) and 17-day-old flies (motor neuron expression). Consistent with our previous findings at 10 days (**Fig. 5d**) or 12 days (**Fig. 5c**), the climbing defects in *hsPEX19-FV*-expressing *Httex1-93Q* flies were significantly rescued compared to their vector controls (see **Supplementary Fig. 14a** and **Figure-response 1**). As a note, we adjusted the climbing assay by increasing the time window from 5 to 6 seconds (15-day-old flies) or 10 seconds (17-day-old flies) to account for the age-related decline in locomotion. While we confirmed that FV also exhibited a suppression effect at 17 days (**Figure-response 1**), we did not include these new results in the revised manuscript due to concerns regarding the potential confounding effects of sick survivors.

Figure-response 1. Climbing ability of 17-day-old adult flies (*Httex1-20Q* and *Httex1-93Q*) expressing vector control, *hsPEX19-WT*, and *hsPEX19-FV*

3.3. Finally, filter trap assays (similar to Fig.2c) should be performed with fly head extracts to show the impact of PEX19-FV on mHTT_{ex1} aggregation in vivo. The results presented in Extended Data Fig. 7 are not fully convincing. The detected puncta may be vesicular structures rather than mHTT_{ex1} aggregates. Additional data are needed to substantiate the relationship between mHTT_{ex1} aggregation and survival of HD transgenic flies.

We thank the reviewer for this suggestion. We performed the filter trap assay using fly head extracts. Consistent with the results in **Supplementary Fig. 13**, the results of the filter trap assay showed that *hsPEX19-FV* significantly suppressed Httex1-93Q aggregation (**Supplementary Fig. 14b, c**), further supporting a correlation between increased mHttex1 aggregation and reduced survival of HD transgenic flies.

Minor points

Fig. 2 (e): Data for FI also fits the results of filter trap assay in Extended Data Fig. 3 (c). Mutation FV is more effective in inhibiting the aggregation than mutation FI.

As the reviewer noticed, all data from the filter trap assay, ThT fluorescence assay, and TEM analysis (**Figs. 2c-f** and **Supplementary Figs. 5** and **6**) showed that *hsPEX19-FV* is more effective in suppressing mHttex1 aggregation than *hsPEX19-FI*. We have edited the manuscript accordingly.

Fig. 3 (d,e): Check legend > *hsPE+X19-FVBpa*? For (d), what is presented by the band below the complex *PEX19Bpax51Q*? Truncated *PEX19*? Pattern matches the pattern of *PEX19* only. How is *PEX19* getting truncated?

Thank you for catching the typo. We have corrected it.

For Bpa incorporation, we inserted the amber stop codon TAG at the F255 residue of *PEX19-FV*, and we coexpressed *PEX19-F255-amber* with a modified aminoacyl-tRNA synthetase, which enables the incorporation of Bpa into the amber stop codon. In the absence of Bpa, protein translation terminates at the amber codon (255th amino acids), generating the truncated *PEX19* protein (**Figure-response 2a**, lane 2). In the presence of Bpa, Bpa can be incorporated at the amber codon, generating a full-length of *PEX19^{Bpa}* (**Figure-response 2a**, lane 3). However, as shown in **Figure-response 2b**, the efficiency of Bpa incorporation is not always 100%. Based on our experience, this efficiency varies depending on amber codon locations in the same gene. More importantly, since this truncated *PEX19* does not have Bpa, it will not crosslink to Httex1-51Q.

Figure-response 2. Expression and purification of PEX19^{Bpa}.

Supplementary Table 1: Not provided.

For the initial submission, Supplementary Table 1 was submitted as a separate Word file, while Extended data figures were attached in the main text file. After reformatting the manuscript according to the guidelines of *Nature Communications*, we put both the Extended data figures and Supplementary Table 1 in the Supplementary Information file.

Extended Data Fig. 1 (c,d,e): What is the unit? Bars do not match blots. Blot shows the opposite result. Were 25Q and 97Q analyzed separately? For (e), which band was analyzed (upper or lower)?

We apologize for any confusion caused by these figures. Since 25Q and 97Q samples were analyzed separately and normalized to their scPEX19-WT sample, we split the previous Extended Data Fig. 1c to form new **Supplementary Fig. 1e** and **Fig. 1f**. In addition, the total PEX19 expression levels were analyzed by summing both PEX19 (upper) and PEX19^F (lower) bands. We have added this information to the figure legends.

Extended Data Fig. 4 (e): The GFP channel for hsPEX19-WT looks similar to the vector control. Is that anticipated?

As shown in **Fig.2g-i**, the Httex1-134Q aggregation level of *hsPEX19-WT* is ~50% of the vector control. The representative image in **Supplementary Fig. 7f** shows that the vector control expressing cells have large Httex1-134Q inclusions. In contrast, in the case of *hsPEX19-WT* expressing cells, one cell has one large Httex1-134Q inclusion, and the other cell has small aggregates (white arrows in **Figure-response 3** (Top)). Although *hsPEX19-WT* expressing cells also have a mixture of small aggregates and soluble proteins (**Figure-response 3** (Middle & Bottom)), collaborating the data in **Fig. 2g-i**, we thought the current **Supplementary Fig. 7f** is a better representative image.

Figure-response 3. Confocal images of HEK293T cells coexpressing Httex1-134Q-GFP and *hsPEX19-WT*. White arrows indicate the aggregated Httex1-134Q-GFP in the cells.

Extended Data Fig. 6 (d,e,f,g): EV is the abbreviation for what?

We apologize for this mistake. EV is the abbreviation for Empty Vector that was used in the previous version of our manuscript. We have now edited EV to “Vector control”.

Reviewer #3 (Remarks to the Author):

The manuscript by Oh et al seeks to engineer PEX19, involved in the targeting of peroxisomal proteins, towards the polyQ expansion-containing mHttex1 proteins involved in Huntington's disease. The authors identified a double mutant of Pex19, using a yeast model of mHttex1 toxicity, which shows significantly improved anti-aggregation activity towards mHttex1 compared to wildtype Pex19. The alteration in activity is transferrable to human PEX19, and mechanistic studies support a new interaction between the engineered PEX19 and the N17 motif in mHttex1 that is responsible for improved activity. The efficacy of the engineered PEX19 was tested in a *Drosophila* model, which showed modest rescues of the toxicity of Httex1-93Q overexpression. The study represents a successful first step in an interesting engineering effort to generate molecular chaperones targeting disease-linked protein aggregation problems. As a rather new approach, it presents some difficulty in evaluation. In particular, I struggled on the bigger context and impact of this study, as it is neither a tour de force engineering effort nor a fundamental study. In the end, the most significant lessons for me are the following: the study demonstrates that it is fairly straight-forward to significantly improve/generate a new chaperone function with only a few mutations, and that by targeting unique motifs in a target protein (such as N17 in mHttex1), it is possible to generate client specificity in the chaperone. I do have a number of reservations on how strong the manuscript demonstrated these points and how effective the chosen approach is, which are detailed below.

1. What is the rationale for choosing Pex19, which has limited interaction with mHtt, as the starting point for engineering, rather than improving on chaperones that naturally have strong activity towards Htt? I think that there is a case against using Hsp104, which is a large yeast protein with limited client specificity and immunogenic potential. However, other mammalian chaperones such as NAC, Hsp70 and J proteins can be candidates. What is the advantage of Pex19 over these alternative options?

Thank you for considering what our study contributes to the field. We do indeed argue that the paper shows that straightforward engineering of a simple ATP-independent membrane protein chaperone might be a rationally designable therapeutic avenue for addressing proteotoxicity for HD. Our workflow could be generalizable to other disorders.

For therapeutic applications, along with enhanced chaperone activity, eliminating side effects caused by other factors that interact with the target chaperone is likely critical. Thus, we aimed to select a chaperone that acts independently. PEX19 does not require any cochaperone, cofactors, or self-assembly process for optimal chaperone activity. However, the promiscuous Hsp70s interact with J proteins, NEFs, and co-factors (ATP) for the regulation of their chaperone activity and substrate specificity (PMID: 20651708, PMID: 31253954). Furthermore, in the case of NAC, the NAC α and NAC β subunits work together on mHttex1 aggregates (PMID: 30982745). Based on our experience in bacteria and mammals, exogenous expression of two different proteins does not always produce equal amounts of proteins in the cells. This raises the concern that one protein

in excess might affect other cellular mechanisms, thereby introducing complications for therapeutic treatment.

As discussed in comment *Reviewer 1.18*, peroxisomal dysfunction observed in peroxisome biogenesis disorders is linked to neurodegenerative diseases. Furthermore, a previous study showed that *PEX19^{-/-}* adult flies exhibit a defect in climbing ability, suggesting the potential functional relevance of PEX19 to neurodegeneration (PMID: 29282281). In addition, since PEX19 acts independently for its activity, we hypothesized that PEX19 would be an ideal candidate for protein engineering to reduce mHttex1 aggregation.

2. Did the engineered mutations interfere with peroxisome targeting? Does overexpression of the mutant Pex19 interfere with wildtype Pex19 function? These are confusing points throughout the manuscript. Farnesylation is important for Pex19 function, and overexpression of the engineered Pex19 drastically reduced Pex19 farnesylation levels. One can infer from this that either the mutants are not farnesylated (and therefore not active in PMP targeting), or that their expression inhibits Pex19 farnesylation. However, the authors assert that the mutants is unaltered in mediate PMP targeting, but the only evidence is that unfarnesylated mutant Pex19 co-purifies with the substrate Pex26. Clarifying this point is important in assessing the approach and the activity/specificity of engineered chaperone mutants.

We apologize for any confusion caused by farnesylation status. Although we did not report this in the previous version of the manuscript, in contrast to *scPEX19* variants, we consistently noticed that a majority of *hsPEX19* variants were farnesylated in HEK293T cells. In the previously submitted Extended Data Fig 4a, we also observed non-farnesylated PEX19 bands (see **Figure-response 4** below). To clearly visualize the non-farnesylated band, we repeated the Western blot analysis with a larger amount of the same cell lysates. As shown in **Supplementary Fig. 7a and d**, similar to *hsPEX19*-WT, over 95% of *hsPEX19* variants were farnesylated in HEK293T cells. Potentially due to different farnesyltransferases in yeast and human (PMID: 8494894), the farnesylation states of *scPEX19* and *hsPEX19* variants might be different.

Furthermore, PEX19 is known to interact with another peroxisomal membrane protein, PMP70 (also served as a peroxisome marker) (PMID: 14709540). Along with the PEX19 pulldown assay with PEX26, we also showed that overexpression of *hsPEX19* variants did not alter the overall peroxisomal integrity of PMP70 (**Supplementary Fig. 7e, f**).

Figure-response 4. Farnesylation levels of *hsPEX19* proteins in HEK293 cells. Overexposed image of Extended Data Fig. 4a showed that exogenously expressed *hsPEX19* variants were largely farnesylated.

3. Most amyloid diseases are strongly seed-catalyzed, such as amyloid beta in Alzheimer's and alpha-synuclein in Parkinson's. Is the aggregation of mHtt also seed-catalyzed? If so, does the engineered Pex19 protect against seeded aggregation of mHtt, which would be closer to conditions in a diseased patient?

We thank the reviewer for this suggestion. Similar to other amyloid diseases, mHttex1 aggregation is known to be seed-catalyzed (PMID: 30193095, PMID: 29282287). We also observed that the inclusion of ~3% seeds (100 nM) in the aggregation reaction stimulates fibril formation of Httex1-51Q (**Supplementary Fig. 6a**). Even in the presence of 100 nM seed, PEX19 variants completely suppressed fibril formation at 36 h (**Supplementary Fig. 6b**). Thus, our data suggest that the engineered PEX19 is capable of protecting the seed-catalyzed aggregation of mHttex1.

4. How does the efficacy of the engineered Pex19 compare with native mammalian chaperones that have been reported to antagonize mHtt aggregation, such as NAC and DNAJB6, in vitro and in the *Drosophila* model? In particular, how does the effect of overexpressing these chaperones compare with expressing mutant Pex19? A higher activity of the mutant Pex19 will strongly support the engineering approach.

Using the filter trap assay, we first compared the chaperone activity of *hsPEX19-FV* with NAC. As shown in **Supplementary Fig. 4e, f**, *hsPEX19-FV* exhibited much stronger suppression activity of Httex1 aggregation than NAC.

In addition, previous studies showed that overexpression of *hsDNAJB1* significantly reduced mHttex1 aggregation in various mammalian cells, whereas *hsHSPA1A* did not exhibit the suppression activity for mHttex1 (PMID: 35948542, PMID: 17822698). Thus, we further assessed the effect of *hsPEX19-FV* on climbing ability using *hsDNAJB1* and *hsHSPA1A* (as a control) with commercially available chaperone transgenic lines. We additionally confirmed the neuronal expression of these chaperones in the transgenic flies (**Figure-response 5**). As shown in **Supplementary Fig. 15a**, *hsPEX19-FV* drastically rescued the climbing ability of 15-day-old *Httex1-93Q* flies, whereas *hsDNAJB1* and *hsHSPA1A* did not display any significant rescue effect compared to the vector control. Furthermore, in contrast to *hsPEX19-FV*, neither *hsDNAJB1* nor *hsHSPA1A* reduced the Httex1-93Q aggregates (**Supplementary Fig. 15b, c**). Collectively, our data suggest that *hsPEX19-FV* has a stronger suppression activity for mHttex1 aggregation than the tested chaperones, thereby more effectively reducing mHttex1-induced neurotoxicity.

Figure-response 6. Immunohistochemistry staining of chaperones in *hsPEX19-FV*, *hsDNAJB1*, and *hsHSPA1A* *Drosophila* lines.

5. For the primary neuron studies (Fig. 5A):

- At what level is PEX19 overexpressed compared to native levels?

We tried to detect the protein expression levels of PEX19 in the primary neuron multiple times before and after submission. As shown in **Figure-response 6a**, we observed very faint bands representing the endogenous PEX19 protein. Since intensity measurement in the Western blot has a certain linear range and the endogenous PEX19 band is out of this linear range and near the detection limit, we think quantification would not work properly.

- The neurons look abnormal when PEX19 is overexpressed. The nuclear area appears much smaller (was the map2 channel saturated?). The images appear to be an overcrowded area with too many overlapping neurons. The authors should provide clearer images, preferably with quantification of the axon morphology.

To image the fragmented neurons more clearly, we often used high laser power for confocal microscopy, resulting in a saturated MAP2 channel. As shown in **Figure-response 6b**, our additional data suggests that overexpression of *hsPEX19-FV* is unlikely to affect the size of the nuclear area in the primary neurons. In the revised manuscript, we used another neuronal marker, Tuj1, and quantified the degree of fragmentation of isolated neurons (**Fig. 5a, b**).

Figure-response. 6. Western blot images (a) and confocal microscopy images (b) of mouse striatal neurons coexpressing mHttex1 and *hsPEX19*.

6. For the *Drosophila* studies:

- The aggregation of the Httex1 proteins from these neurons should also be analyzed.

We also thank the reviewer for this suggestion. Please refer to our response above (*Reviewer 2-3.3*). We have added the data to **Supplementary Fig. 14b, c** as well as **Supplementary Fig. 15b, c**, and edited the manuscript accordingly.

- It appears that overexpression of PEX19 (WT and mutant) decreases the lifespan of otherwise healthy flies, but this point was skipped over.

Following *Reviewer 2's* suggestion, we now include the complete survival curves for both the *W1118* and *Httex1-20Q* lines (please refer to our response for the comment *Reviewer 2-3.1* and also see below). Based on statistical analysis, we found that overexpression of PEX19-WT significantly increased the lifespan of *Httex1-20Q* (**Fig. 5f**). We added this finding to the main text (page 13).

- I don't understand the statistics: in 5E, the difference between flies with and without PEX19 overexpression is pretty large (a horizontal difference of ~ 8 days), but was labeled * or ns. In 5F, the difference is at most two days, but was given ****.

We apologize for any confusion caused by statistical analysis with incomplete survival curves. We realized that assessing the statistical significance of the data with incomplete survival curves would not be valid. Thus, we used both log-rank and Gehan-Wilcoxon tests to evaluate the data significance of complete survival curves (**Fig. 5e-g**, also refer to our response for *Reviewer 2-3.1*). These tests compare the contributory differences between vector control and PEX19 on the same days. As an example, ~80% of 93Q-FV flies survived on day 17, whereas ~30% of 93Q-vector control flies survived on the same day. In contrast, ~40% of *W¹¹¹⁸*-FV flies survived on day 50, whereas ~30% of *W¹¹¹⁸*-vector control flies survived. Thus, the rescue effect of *hsPEX19-FV* in *Httex1-93Q* flies is much larger than in healthy flies at a later time. Based on the statistical analysis, we have edited the manuscript accordingly (page 13).

7. An important point of this study is that the mutant Pex19 is specific for mHttex1 Huntington, based on comparison with TDP43 and another polyQ protein. The specificity will be more convincingly demonstrated from an IP-MS approach to test if the engineered and overexpressed PEX19 show enhanced interaction with other cellular proteins besides mHttEx1.

Membrane protein chaperones, including PEX19, recognize their substrates based on the hydrophobicity of the transmembrane domains. However, the IP-MS approach requires extensive washing with detergents to eliminate non-specifically bound proteins. In our preliminary data, as predicted, when we included 0.01% digitonin during GFP-trap IP, PEX19 proteins were in the elution fractions (**Figure-response 7a**). In this case, a majority of proteins were in the elution fractions due to inefficient washing (**Figure-response 7b**). When 0.2% digitonin was included, all PEX19 proteins were in the flowthrough fractions (**Figure-response 7a**). Subtle changes in detergent concentration would generate biased results toward a certain range of hydrophobic substrates, and thus the IP-MS approach would not be suitable to globally identify PEX19 substrates.

To further support the substrate specificity of *hsPEX19* variants, we additionally mutated the residues in the N17 domain and tested their interaction with these substrates (**Fig. 4a, d**). The results of our Bpa crosslinking assay suggest that the $\alpha 4$ helix of PEX19 specifically recognizes the N17-like hydrophobic domain (**Fig.4a, d**).

Figure-response 7. Immunoprecipitation (IP) analysis of *hsPEX19* binding to 134Q-GFP. IP experiments with HEK293T cell lysates were performed using GFP-Trap beads. Each fractions were analyzed using Western blot (a) and Coomassie staining (b). The samples in the Elution fractions were loaded 5~10-fold excess than other fractions.

Reviewer #1 (Remarks to the Author):

I think the authors did a very nice job of addressing my concerns. With the newly added data and analyses, I think the manuscript is improved and appropriate for publication in Nature Communications.

Thank you again for all the comments from the reviewer. The comments were really helpful in improving our manuscript.

Reviewer #2 (Remarks to the Author):

Oh et al/NCOMMS-24-26029A

Engineering a membrane protein chaperone to ameliorate the proteotoxicity of mutant huntingtin

The authors have now comprehensively addressed my concerns. The paper is now suitable for publication in Nature Comm. I have no further comments. This is a well written paper that is of high importance for the scientific community working on HD.

Thank you for all the constructive comments from the reviewer and for acknowledging what our study contributes to the field.

Reviewer #3 (Remarks to the Author):

The authors have addressed most of my questions on the previous version of the manuscript. Before acceptance, though, the authors need to address a few minor issues:

We greatly appreciate the reviewer's comments and suggestions.

1. Supplementary figure 6: The effect of seed in accelerating HttQ aggregation is very modest, and the aggregation observed in the presence of 100 nM seed is likely / probably largely driven by unseeded reaction. The author should tone down the conclusion that the chaperone prevents seed-catalyzed Htt aggregation.

Although the fluorescence intensity values of 100 nM seeded samples were increased up to 30-40% at 36 h compared to the control, we agree with the reviewer that the effect of 100 nM in the initial Httex1-51Q aggregation kinetics is very modest. Thus, we rephrased the sentence as "the PEX19 variants exhibit the potential to reduce the seeded aggregation of mHttex1".

2. Supplementary figure 8: In the SPR data, neither the association nor dissociation time traces have reached completion to accurately determine the rate constants. The dissociation trace is particularly problematic (k_{off} is $10^{-4} s^{-1}$, or $\tau = 10000$ s), but the trace only lasted 600 s). In fact, if one uses the magnitude of binding at each concentration as a proxy of equilibrium, the binding has not fully saturated at 100-250 nM. I think the SPR

data provide evidence for high nanomolar to sub-micromolar affinity binding to HttQ51 that is specific for the N17 motif, but the K_d of 2.5 nM is likely incorrect. The authors should rephrase the conclusion to reflect the uncertainty.

We totally agree with the reviewer that the kinetic analysis could be inaccurate due to the incomplete association and dissociation curves in the BLI experiments. More importantly, due to N-terminal exposure of the N17 domain, the actual binding affinity of Httex1-51Q-WT to *hsPEX19-FV* could be different from GST-tagged Httex1-51Q-WT (used in the BLI experiments). Thus, in the revised manuscript, we only used the data to support the conclusion that the N17 domain of Httex1-51Q is a primary binding domain for *hsPEX19-FV* and edited the corresponding sentence accordingly; “Consistent with this, GST-Httex1-51Q-WT readily bound to *hsPEX19-FV*, whereas no binding was observed with GST-Httex1-51Q- Δ N (Supplementary Fig.8a, b)”. Furthermore, we have edited the figure legends of Supplementary Fig. 8 to further add the uncertainty of the kinetic analysis.

3. Line 318: reference to Fig. 4 i-k for PEX19 xlink to PEX26. Figures 4j and 4k are with Htt51Q, not PEX26.

Thank you for catching the mistake. Since both Fig. 4i and 4k show the PEX19-PEX26 crosslinks with two different antibodies (4i for PEX19 and 4k for PEX26 with the same samples), we corrected it as “Fig. 4i and k”.